# ⚒ PHYSTOOLBENCH: BENCHMARKING PHYSICAL TOOL UNDERSTANDING FOR MLLMS

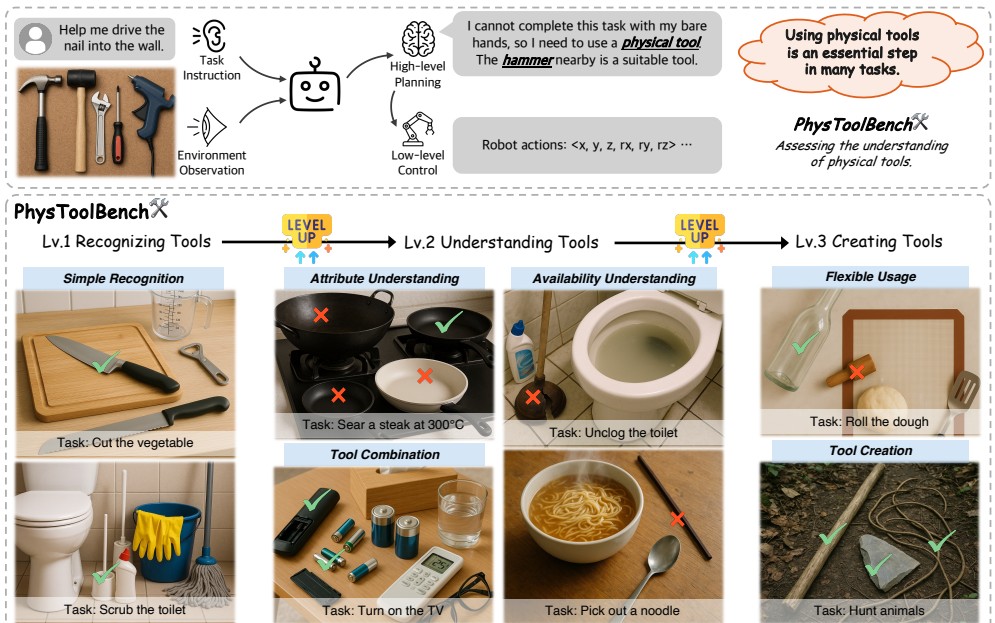

Figure 1: For an Embodied Agent, using physical tools is crucial in many tasks. The understanding of physical tools significantly impacts the task's success rate and execution efficiency (Top). PhysToolBench (Bottom) systematically evaluates the understanding of physical tools of multimodal LLMs. The benchmark is designed with three progressive levels of difficulty and employs a Visual Question Answering (VQA) format. Notice that in the actual benchmark, tools in the images are numerically labeled. Images here are for illustrative purposes only.

## ABSTRACT

The ability to use, understand, and create tools is a hallmark of human intelligence, enabling sophisticated interaction with the physical world. For any general-purpose intelligent agent to achieve true versatility, it must also master these fundamental skills. While modern Multimodal Large Language Models (MLLMs) leverage their extensive common knowledge for high-level planning in embodied AI and in downstream Vision-Language-Action (VLA) models, the extent of their true understanding of physical tools remains unquantified. To bridge this gap, we present **PhysToolBench**, the first benchmark dedicated to evaluating the comprehension of physical tools by MLLMs. Our benchmark is structured as a Visual Question Answering (VQA) dataset comprising over 1,000 image-text pairs. It assesses capabilities across three distinct difficulty levels: *1)* Tool Recognition: Requiring the recognition of a tool's primary function. *2)* Tool Understanding: Testing the ability to grasp the underlying principles of a tool's operation. *3)* Tool Creation: Challenging the model to fashion a new tool from surrounding objects when conventional options are unavailable. Our comprehensive evaluation of 32 MLLMs—spanning proprietary, open-source, specialized embodied, and backbones in VLAs—reveals a significant deficiency in the tool understanding. Furthermore, we provide an in-depth analysis and propose preliminary solutions. Code and dataset are publicly available[1].

---

[1]https://github.com/PhysToolBench/PhysToolBench

# 1 INTRODUCTION

*Man is a tool-using animal. Without tools, he is nothing; with tools, he is all.*

—Thomas Carlyle

A key factor in humanity's success throughout natural evolution is the ability to create and utilize a vast array of tools to enhance survival and prosperity. With the advancement of technology, humans continuously reshape the physical world, inventing diverse instruments to extend the boundaries of their capabilities. For an embodied intelligent agent designed to complete physical tasks, the use of tools is a prerequisite for achieving success and efficiency. For instance, as illustrated in Fig. 1 (Top), a robot must use a hammer to drive a nail into a wall—a task it cannot accomplish with its bare manipulators. Arguably, a profound understanding of physical tools is a fundamental precondition for Artificial General Intelligence (AGI).

Multimodal Large Language Models (MLLMs) (Bai et al., 2025; Hurst et al., 2024; OpenAI, 2025b; Anthropic, 2025; Comanici et al., 2025), which can process inputs from both vision and language modalities, have acquired substantial common-sense knowledge from being trained on massive datasets. They show great promise for evolving into AGI and have been the focus of numerous studies for deployment in robotics. Some studies employ MLLMs as high-level planners (Yuan et al., 2025; Team et al., 2025a; Driess et al., 2023), while others utilize them for low-level control as the backbone of Vision-Language-Action (VLA) models (Black et al., 2024; Kim et al., 2024; Wen et al., 2025; Black et al.). In either case, interaction with the physical world is fundamental, which inevitably involves the use of physical tools. Although some research has demonstrated that MLLMs possess a preliminary understanding of tools (Gao et al., 2025; Tang et al., 2025; Trupin et al., 2025), the true depth of physical tool comprehension remains largely unexplored.

Based on these considerations, we propose **PhysToolBench**, a benchmark for evaluating an agent's understanding of physical tools. To the best of our knowledge, this is the first benchmark specifically designed for this purpose. To evaluate an agent's practical capabilities, we designed a Visual Question Answering (VQA) benchmark that simulates a robotic workflow. Presented with a task and an image of objects, the agent must select the appropriate tool(s). As shown in Fig. 1 (Bottom), the benchmark features three difficulty levels to progressively assess the agent's depth of understanding: *1) Easy (Recognizing Tools)*. This fundamental level assesses whether an agent can identify a conventional tool and its primary function. *2) Medium (Understanding Tools)*. This intermediate level probes the agent's comprehension through three distinct challenges: optimal tool selection from functionally similar options, selection of all tools required for a multi-tool task, and assessment of a tool's operational viability based on its physical state. *3) Hard (Creating Tools)*. This advanced level evaluates an agent's inventive capabilities. Faced with a task and no standard tools, the agent must fashion a solution by repurposing or combining available objects, which requires an understanding of the physical principles underlying the required tool.

We evaluate the performance of 32 MLLMs on PhysToolBench, spanning four distinct classes: general-purpose proprietary MLLMs, general-purpose open-source MLLMs, MLLMs tailored for embodied AI, and those functioning as backbones in VLAs. The results demonstrate a clear performance ceiling, with even the most advanced proprietary models scoring no higher than 63%, revealing a profound disparity with human proficiency in tool understanding (over 90%). Furthermore, our analysis uncovers several critical weaknesses in current MLLMs: (1) a failure of small MLLMs, including those within VLA models, to exhibit an emergent ability of tool understanding; (2) a long-tail distribution issue in recognizing and understanding a wide array of tools; (3) a tendency to hallucinate tool affordances and their availability; and (4) inadequate visual reasoning skills. We further propose a "deep visual reasoning" framework to bolster the visual reasoning of MLLM agents. We hope our work will inspire future research on physical tool understanding.

# 2 RELATED WORKS

## 2.1 MLLM AND ITS APPLICATION IN EMBODIED AI

Recent years have witnessed remarkable advancements in Multimodal Large Language Models (MLLMs). Building on the significant success of Large Language Models (LLMs), these models effectively process visual information by leveraging modality alignment techniques (Li et al.,

2022; Radford et al., 2021). Typically, a visual encoder and a connector are employed to link visual data to the LLM, enabling reasoning at the language level and granting Vision-Language Models (VLMs) sophisticated image comprehension capabilities. To date, numerous impressive MLLMs have emerged, including proprietary models (Hurst et al., 2024; OpenAI, 2025b; Comanici et al., 2025; Anthropic, 2025; xAI, 2025), as well as open-source alternatives (Bai et al., 2025; Zhu et al., 2025; Wang et al., 2025b; Wu et al., 2024; Beyer et al., 2024; Wu et al., 2024; Lu et al., 2025). These models have demonstrated powerful visual understanding across a diverse range of tasks.

Beyond general-purpose domains, MLLMs are also finding significant applications in embodied intelligence. On one hand, they are being utilized as the high-level "brain" for task planning in embodied agents, as exemplified by PaLM-E (Driess et al., 2023), RoboBrain (Team et al., 2025a), and Embodied-R1 (Yuan et al., 2025). On the other hand, research has also capitalized on the inherent common-sense knowledge within MLLMs. By adding an action head and fine-tuning on robotic data, they can be transformed into end-to-end Vision-Language-Action (VLA) models capable of directly outputting robot actions. Notable examples of this approach include $\pi_0$ (Black et al., 2024), $\pi_{0.5}$ (Black et al.), and OpenVLA (Kim et al., 2024).

## 2.2 Physical Tool Use in Embodied AI

These advancements in foundation models have empowered robots with the ability to perform fundamental tasks when these models are embodied. For instance, embodied models such as $\pi_0$, $\pi_{0.5}$, and OpenVLA can successfully accomplish basic household chores like folding clothes and tidying desktops. However, while these tasks can be efficiently completed using only the robot's own manipulators, many higher-level, real-world tasks are difficult and even impossible to achieve with robot manipulators alone. Consequently, teaching robots how to use tools to effectively accomplish complex objectives is of critical importance.

Initial research has begun to explore endowing robots with tool-using capabilities. For example, VLMgineer (Gao et al., 2025) employs a VLM agent to assist robots in crafting simple tools to complete tasks. Similarly, Trupin et al. (2025) leverages vision foundation models to enable tool use during task planning. MimicFunc (Tang et al., 2025) establishes an imitation learning framework that allows robots to learn tool manipulation by observing human demonstration videos. Leveraging the common-sense knowledge inherent in MLLMs, these approaches have demonstrated a rudimentary ability to use physical tools. Nevertheless, the depth of physical tool understanding that their "brain"—the MLLM—possesses remains largely unexplored. The primary motivation for our work is to clarify this question by creating a benchmark designed specifically to evaluate the understanding of physical tools within Multimodal Large Language Models.

## 2.3 Related Benchmarks

For Large Language Models (LLMs), a multitude of benchmarks (Wang et al., 2024a; Huang et al., 2023; 2024; Lu et al., 2024a; Ye et al., 2025) have been developed to evaluate their ability to utilize digital tools, such as search engines, translation services, booking systems, etc.. These benchmarks have catalyzed the rapid development of modern LLM Agents, equipping them with the capability to invoke external APIs to accomplish complex tasks. However, a significant gap exists when it comes to physical tools, as there is currently no corresponding benchmark for MLLMs. We argue that such a benchmark is crucial for advancing MLLMs toward becoming true Embodied Agents capable of meaningful interaction with the physical world.

Among existing benchmarks for MLLMs, A4Bench (Wang et al., 2025a) is the most relevant to our research. It operates in a VQA format, presenting an image of a tool and asking the MLLM to identify its function from a set of multiple-choice options. While this can, to some extent, reflect the MLLM's understanding of object affordances, we contend that this question-answering format lacks practical applicability. Our work, therefore, aims to establish a more application-oriented evaluation. We provide the MLLM with a specific task requirement and an image containing several tools, compelling it to answer the question based on the observation. This approach more rigorously assesses whether the MLLM can apply genuine knowledge and reasoning to find the optimal tool, rather than merely relying on the rote memorization of tool-function associations.

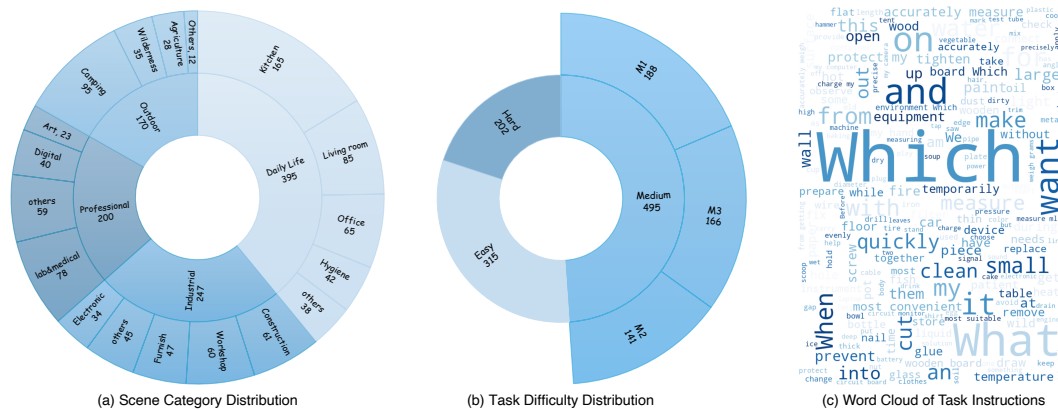

(a) Scene Category Distribution     (b) Task Difficulty Distribution     (c) Word Cloud of Task Instructions

Figure 2: Statistics of PhysToolBench . (a) is the distribution of the category. (b) is the distribution of the difficulty level. (c) is the word cloud of the task description given to MLLMs.

## 3 THE PHYSTOOLBENCH

### 3.1 OVERVIEW

PhysToolBench is a VQA benchmark comprising over 1,000 text-image pairs designed to evaluate an MLLM's understanding of physical tools. Each pair consists of a text prompt outlining a specific task and a corresponding $1024 \times 1024$ image displaying several numerically labeled tools and objects. A core design constraint is that the MLLM is explicitly instructed that the items depicted in the image are the only available things, simulating a realistic robotics scenario with limited resources. The MLLM's objective is to analyze the task and visual information, then output the numerical label(s) of the required tool(s), or "None" if no suitable tool is available. PhysToolBench spans four major domains: Daily Life, Industrial, Outdoor Activities, Professional Settings, and three difficulty levels: Easy, Medium, Hard. Detailed statistics are shown in Fig. 2.

### 3.2 DESIGN PRINCIPLES

To progressively evaluate the depth of an MLLM's understanding, we designed PhysToolBench with three distinct difficulty levels: Easy, Medium, and Hard, each demanding a more profound comprehension of tool properties and functionality.

The **Easy** level assesses fundamental tool recognition. Questions are answerable with basic tool identification and common-sense knowledge. Task prompts are straightforward, and the image always contains a tool whose primary function directly matches the task. For example, to "cut vegetables," the image will include a kitchen knife. The **Medium** level requires a deeper understanding of tools, necessitating reasoning based on specific task constraints. This tier is subdivided into three challenges: 1) *M.1. Attribute Understanding*, requiring comprehension of a tool's specific attributes (e.g., selecting a cast-iron skillet for its high heat tolerance); 2) *M.2. Tool Combination*, evaluating the ability to combine tools to unlock new affordances (e.g., inserting batteries into a remote); and 3) *M.3. Availability Understanding*, testing the recognition of non-functional tools (e.g., identifying a cracked plunger as unusable). The **Hard** level assesses higher-order reasoning and creativity. The model must work backwards from task requirements to innovatively utilize surrounding objects. For instance, if tasked to "tighten a flat-head screw" without a suitable screwdriver, the MLLM must identify that a coin can serve as a substitute.

We propose these difficulty levels as a tiered evaluation standard. The 'Easy' score serves as a prerequisite for basic tool-use planning, 'Medium' benchmarks potential in complex scenarios, and 'Hard' presents a forward-looking challenge for AGI research.

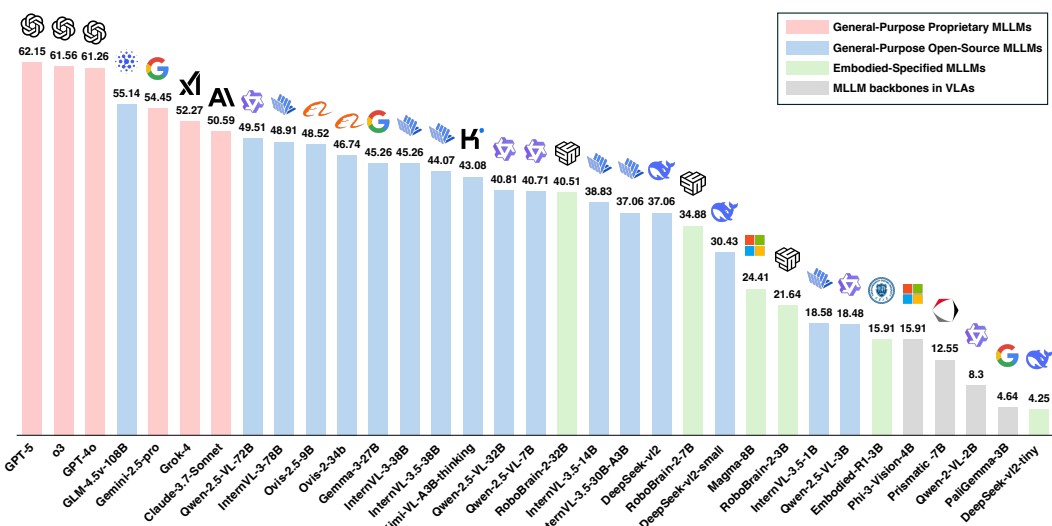

Figure 3: MLLM Leaderboard on our PhysToolBench, ranked by overall performance.

## 3.3 DATASET COLLECTION PROCESS

The collection of test samples for PhysToolBench was conducted in three phases to ensure quality. **Phase 1: Conceptualization.** Human experts designed task-scene pairs, consisting of a task requirement and a detailed scene description, meticulously aligning each scenario with our Easy, Medium, and Hard difficulty criteria. **Phase 2: Image Generation.** Scene descriptions were transformed into visual images primarily using GPT-4o-image (OpenAI, 2025a)(approximately 90%), a process closely supervised by human experts who vetted for quality and realism. For complex objects that the generative model struggled with, we resorted to physical staging and photography(approximately 10%). **Phase 3: Annotation and Verification.** Experts used a custom software tool to apply numerical labels to objects in each image. The entire dataset then underwent a final, thorough review and revision by a separate team to verify its integrity and ensure reliability. More details are provided in Appendix. B.

## 4 EXPERIMENTS ON PHYSTOOLBENCH

### 4.1 BENCHMARK CANDIDATES

We conducted a comprehensive evaluation across four distinct categories of state-of-the-art Multimodal Large Language Models (MLLMs), encompassing **32 models** in total: **a) General-Purpose Proprietary MLLMs:** GPT-5 (2025-08-17) (OpenAI, 2025b), o3 (2025-04-16) (OpenAI, 2025c), ChatGPT-4o-latest (2025-01-29) (Hurst et al., 2024), Claude-3-7-Sonnet-thinking (Anthropic, 2025), Gemini-2.5-pro (2025-05-06) (Comanici et al., 2025), Grok-4 (xAI, 2025). **b) General-Purpose Open-Source MLLMs:** Qwen-2.5-VL-72B-Instruct (Bai et al., 2025), Qwen-2.5-VL-32B-Instruct, Qwen-2.5-VL-7B-Instruct, Qwen-2.5-VL-3B-Instruct, InternVL-3.5-38B (Wang et al., 2025b), InternVL-3.5-30B-A3B, InternVL-3.5-14B, InternVL-3.5-1B, InternVL-3-78B (Zhu et al., 2025), InternVL-3-38B, GLM-4.5V-108B (Team, 2025), Ovis-2-34B (Lu et al., 2024b), Ovis-2.5-9B (Lu et al., 2025), DeepSeek-VL-2 (Wu et al., 2024), DeepSeek-VL-2-small, DeepSeek-VL-2-tiny, Kimi-VL-A3B-thinking-2506 (Team et al., 2025b). **c) Embodied-Specific MLLMs:** RoboBrain-2-32B (Team et al., 2025a), RoboBrain-2-7B, RoboBrain-2-3B, Embodied-R1-3B (Yuan et al., 2025), Magma-8B (Yang et al., 2025) **d) MLLM Backbones of Vision-Language-Action (VLA) models:** Prismatic-7B (Karamcheti et al., 2024) in OpenVLA (Kim et al., 2024), PaliGemma-3B (Beyer et al., 2024) in $\pi_0$ (Black et al., 2024), Qwen-2-VL-2B (Wang et al., 2024b) in DexVLA (Wen et al., 2025), Phi-3-Vision-4B (Abdin et al., 2024) in TraceVLA (Zheng et al., 2024).

The first category of proprietary models was evaluated via their respective APIs. For the latter three categories, the models were downloaded and deployed locally for testing. To ensure a fair comparison, we used a consistent text prompt for all models. The system prompt was designed to encourage

Table 1: Benchmark results on the PhysToolBench. For each difficulty level and scene category, the best performance was marked in **bold** and the second best was marked underline. *Prismatic-7B achieves an unusually high score on the Medium-M3 difficulty. Upon inspecting its reasoning process, we discovered that the model does not generate sound reasoning but instead exhibits a strong tendency to output "None" in all case.

| Categories | Difficulty Level | | | | | Scene Category | | | | Overall↑ |
|---|---|---|---|---|---|---|---|---|---|---|
| MLLM | Easy↑ | m1↑ | m2↑ | m3↑ | Hard↑ | Professional↑ | Industrial↑ | Outdoor↑ | Daily↑ | |
| HUMAN(BEST) | 96.19% | 93.61% | 90.78% | 93.97% | 89.10% | 87.5% | 93.52% | 91.17% | 96.71% | 93.19% |
| HUMAN(WORST) | 91.74% | 87.77% | 85.11% | 90.36% | 81.68% | 80.5% | 85.02% | 87.65% | 93.42% | 87.85% |
| **General-Purpose Proprietary MLLMs:** | | | | | | | | | | |
| GEMINI-2.5-PRO | 78.10% | 48.40% | 46.10% | 45.78% | 36.14% | 58.5% | 61.54% | 46.47% | 51.39% | 54.45% |
| O3 | **93.02%** | 67.02% | 46.81% | 22.89% | **49.50%** | 64.0% | **68.02%** | **61.18%** | 56.46% | 61.56% |
| GPT-4O | 86.03% | **70.74%** | 48.23% | 35.54% | 44.06% | 62.5% | 63.97% | 59.41% | **59.75%** | 61.26% |
| GPT-5 | 90.16% | 63.83% | **50.35%** | 36.75% | 46.04% | **67.5%** | 66.8% | 58.82% | 57.97% | **62.15%** |
| GROK-4 | 73.65% | 46.28% | 30.50% | **52.41%** | 39.60% | 50.5% | 59.92% | 43.53% | 52.15% | 52.27% |
| CLAUDE-3-7-SONNET-THINKING | 74.60% | 58.51% | 35.46% | 27.11% | 35.64% | 53.5% | 55.87% | 45.88% | 47.85% | 50.59% |
| **General-Purpose Open-Source MLLMs:** | | | | | | | | | | |
| QWEN-2.5-VL-72B | 75.56% | 55.85% | 35.46% | 31.93% | 27.23% | 51.5% | 55.47% | 44.71% | 46.84% | 49.51% |
| QWEN-2.5-VL-32B | 67.62% | 43.09% | 30.5% | 22.29% | 19.31% | 42.0% | 49.39% | 37.06% | 36.46% | 40.81% |
| QWEN-2.5-VL-7B | 71.43% | 51.6% | 20.57% | 21.08% | 12.87% | 44.0% | 49.39% | 38.24% | 34.68% | 40.71% |
| QWEN-2.5-VL-3B | 36.51% | 10.64% | 6.38% | 13.86% | 9.9% | 21.5% | 21.46% | 15.88% | 16.2% | 18.48% |
| GLM-4.5V-108B | 90.48% | 65.43% | 36.88% | 16.27% | 35.15% | 62.5% | 59.92% | 56.47% | 47.85% | 55.14% |
| GEMMA-3-27B | 68.57% | 57.45% | 31.91% | 19.88% | 27.72% | 50.0% | 48.99% | 42.94% | 41.52% | 45.26% |
| INTERNVL-3.5-38B | 70.79% | 50.53% | 29.08% | 18.67% | 27.72% | 51.0% | 49.8% | 37.65% | 39.75% | 44.07% |
| INTERNVL-3.5-30B-A3B | 66.03% | 37.77% | 20.57% | 15.06% | 20.79% | 41.0% | 43.32% | 31.18% | 33.67% | 37.06% |
| INTERNVL-3.5-14B | 66.03% | 40.43% | 21.99% | 21.08% | 21.29% | 44.0% | 44.94% | 31.76% | 35.44% | 38.83% |
| INTERNVL-3.5-1B | 38.73% | 19.68% | 4.26% | 3.61% | 8.42% | 22.5% | 18.22% | 20.0% | 16.2% | 18.58% |
| INTERNVL-3-78B | 79.05% | 53.72% | 39.01% | 21.08% | 27.23% | 52.0% | 56.28% | 42.94% | 45.32% | 48.91% |
| INTERNVL-3-38B | 77.78% | 44.68% | 31.91% | 16.87% | 27.72% | 51.0% | 53.04% | 41.18% | 39.24% | 45.26% |
| OVIS-2.5-9B | 80.63% | 55.85% | 42.55% | 17.47% | 21.29% | 57.0% | 56.28% | 44.12% | 41.27% | 48.52% |
| OVIS-2-34B | 83.17% | 45.21% | 35.46% | 15.66% | 24.75% | 56.5% | 52.23% | 40.0% | 41.27% | 46.74% |
| DEEPSEEK-VL2-27B | 71.75% | 39.89% | 19.86% | 6.63% | 17.33% | 44.0% | 42.91% | 35.88% | 30.38% | 37.06% |
| DEEPSEEK-VL2-SMALL-16B | 64.44% | 28.19% | 10.64% | 10.24% | 9.9% | 36.0% | 37.65% | 25.88% | 25.06% | 30.43% |
| DEEPSEEK-VL2-TINY-3B | 7.62% | 2.13% | 2.84% | 4.22% | 1.98% | 7.0% | 5.26% | 2.94% | 2.78% | 4.25% |
| KIMI-VL-30B-A3B-THINKING | 79.05% | 45.21% | 31.21% | 18.67% | 13.37% | 46.5% | 48.58% | 40.59% | 38.99% | 43.08% |
| **Embodied-Specified MLLMs:** | | | | | | | | | | |
| ROBOBRAIN-2-32B | 75.87% | 49.47% | 19.86% | 6.63% | 19.31% | 48.5% | 47.37% | 39.41% | 32.66% | 40.51% |
| ROBOBRAIN-2-7B | 66.03% | 44.68% | 13.48% | 10.84% | 11.88% | 36.5% | 41.7% | 34.71% | 29.87% | 34.88% |
| ROBOBRAIN-2-3B | 46.35% | 18.62% | 3.55% | 11.45% | 6.93% | 25.5% | 28.74% | 18.24% | 16.71% | 21.64% |
| EMBODIED-R1-3B | 38.41% | 6.38% | 4.96% | 4.22% | 6.93% | 23.0% | 20.24% | 11.76% | 11.39% | 15.91% |
| MAGMA-8B | 46.35% | 29.26% | 0% | 3.01% | 20.3% | 19.0% | 29.55% | 25.88% | 23.29% | 24.41% |
| **MLLM backbones in VLAs:** | | | | | | | | | | |
| PALIGEMMA-3B | 7.94% | 10.11% | 0% | 0% | 1.49% | 6.0% | 4.86% | 4.12% | 4.05% | 4.64% |
| PHI-3-VISION-4B | 33.97% | 12.77% | 4.26% | 3.01% | 9.41% | 20.5% | 19.43% | 11.18% | 13.42% | 15.91% |
| QWEN-2-VL-2B | 19.37% | 1.6% | 0.71% | 7.83% | 2.97% | 7.0% | 9.31% | 4.12% | 10.13% | 8.3% |
| PRISMATIC-7B | 6.98% | 4.26% | 1.42% | *56.02% | 0.99% | 11.0% | 13.77% | 8.24% | 14.43% | 12.55% |

a Chain-of-Thought (Wei et al., 2022), explicitly asking the models to reason before providing their final answer. The only exception was for models that feature a native, built-in "thinking" mode, in which case we allowed them to utilize their default inference process without modification. We also recruited 5 human participants as testers to serve as a reference.

## 4.2 OVERALL RESULTS

As shown in Tab. 1, MLLMs generally underperform, with most scoring below 60%—a result far inferior to human performance, which consistently achieves at least 87.85% overall accuracy. This indicates that contemporary MLLMs have a superficial understanding of tool usage. Among the models evaluated, proprietary general-purpose MLLMs performed best. The OpenAI series (o3, gpt-4o, and gpt-5) all exceeded the 60% threshold, with gpt-5 leading the group. Open-source general-purpose MLLMs followed, typically scoring above 40%. GLM-4.5V was a notable exception, achieving 55.14% and outperforming not only its open-source peers but also some proprietary models, highlighting its significant potential. Embodied-specific MLLMs demonstrated some capability but lagged behind the general-purpose open-source models. Lastly, MLLM backbones within VLA frameworks exhibited the weakest performance, likely due to their limited number of parameters. An overall leaderboard of MLLMs is shown in Fig. 3. We provide a set of complete VQA results in Appendix. C.

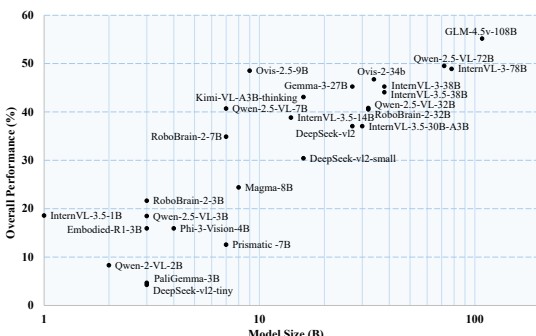

Figure 4: **Overall performance v.s. model size** for open-source MLLMs. A significant correlation is observed between performance and model size.

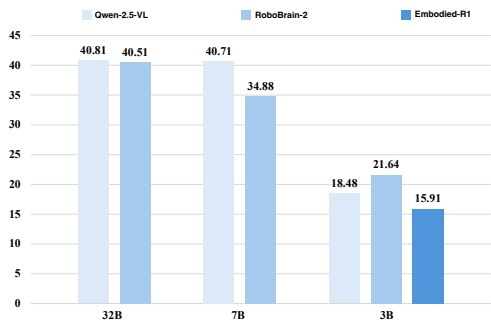

Figure 5: Performance comparison between the embodied models and their base model.

### 4.3 FINDINGS ON PHYSTOOLBENCH

***F.1. A foundational ability to understand tools emerges in large models with sufficient scale.*** As shown in Fig. 4, our evaluation of numerous open-source models reveals that there's a significant correlation between the understanding of physical tool and the size of the model. Furthermore, for the easy difficulty setting in Tab. 1, we also observe that a foundational understanding of tool usage emerges once a model reaches a certain scale, which we preliminarily identify as approximately 10 billion parameters. Most models exceeding this 10B threshold achieve an accuracy of 60-70% on easy-level tasks. In contrast, performance drops significantly for smaller models; those with fewer than 5B parameters generally score below 50% on easy tasks and have an overall accuracy below 25%. Consequently, we recommend selecting MLLMs with more than 10 billion parameters for applications in embodied intelligence.

***F.2. A long-tail problem persists in tool recognition and understanding, even for the most advanced MLLMs.*** Although top-tier MLLMs are proficient at identifying common objects, their performance diminishes for less common items, creating a long-tail effect. A notable finding is the models' pronounced weakness in the subcategory of digital products. They frequently fail to distinguish between visually similar items, such as HDMI versus DP cables and Type-C versus Lightning charging ports. This deficiency is widespread in open-source models, where even the highly capable GLM-4V shows errors in basic recognition, as in Fig. 6. (a). Closed-source models offer a marginal improvement but still demonstrate only a shallow comprehension. As an example in Fig. 6. (c), most top-tier proprietary models do not grasp the functional requirement that a monitor must be connected to a laptop using an HDMI cable and an adapter if the laptop only has a Type-C port.

***F.3. Embodied-specific MLLMs show no significant advantage on PhysToolBench.*** Models specifically fine-tuned for embodied tasks, such as RoboBrain2 and Embodied-R1, do not exhibit a notable performance improvement on our benchmark. RoboBrain2's parameters were initialized from Qwen2.5VL and subsequently fine-tuned on a combination of general vision and robotic datasets. Nevertheless, as shown in Fig. 5, its 32B, 7B variants all performed slightly below their Qwen2.5VL backbone of equivalent scale. A similar trend was observed with Embodied-R1-3B, which, despite being fine-tuned from Qwen-2.5-VL-3B, also achieved a marginally lower score than the original model. These findings indicate that the fine-tuning process did not confer an enhanced understanding of tools. We hypothesize that current robotic datasets may require more high-quality data centered on tool comprehension to advance these models' physical tool understanding.

***F.4. MLLMs exhibit a critical deficiency in comprehending tool availability, failing to grasp the fundamental principles of their utility.*** The M3 difficulty tier of our benchmark was specifically designed to probe this issue by incorporating simple "traps": presenting the correct tool for a task but in a damaged or non-functional state. Counter-intuitively, as shown in Tab. 1, models found this task more difficult than the "Hard" tier, which requires complex reasoning for tool creation. For instance, in the selected four cases in Fig. 6. (d), none of the MLLMs could identify when the tools are unavailable. This outcome strongly suggests that the models' comprehension of tools is shallow and relies on surface-level "common sense" associations rather than a robust understanding of their core functionality, leading them to hallucinate the tool's usability.

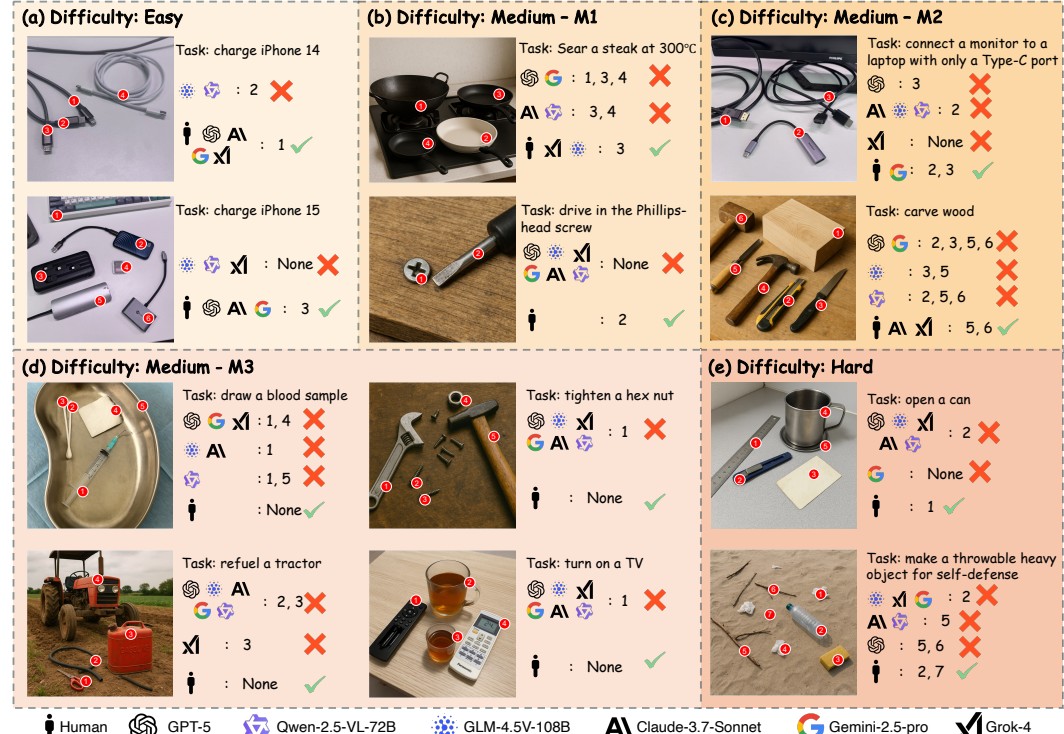

Figure 6: Some results of PhysToolBench. We showcase illustrative examples for each difficulty level, along with the answers from several top-tier models and human participants. Note that the markers are intentionally enlarged for visualization purposes.

The implications of this hallucination for embodied agents are severe. An agent that cannot recognize a tool as non-functional may attempt to use it, resulting in mission failure and significant safety hazards—for instance, fueling a tractor with gasoline, drawing a blood sample with a damaged syringe. We contend that addressing this issue is critical for advancing embodied AI.

***F.5. The MLLM backbones in current VLAs are extremely weak.*** Our evaluation revealed that the MLLM backbones of the contemporary VLA models exhibit exceptionally poor performance on PhysToolBench, with overall scores universally below 15%. This result calls into question the prevailing assumption that VLAs can effectively inherit "common sense" from their base MLLM and then achieve generalization through fine-tuning on robotic action datasets. Our findings suggest that the foundational "common sense" of these MLLMs is profoundly insufficient for general-purpose intelligence. We posit that this fundamental limitation cannot be rectified through fine-tuning on robotic datasets of a modest scale. Consequently, we conclude that advancing the VLA paradigm will require a two-pronged approach: first, leveraging significantly larger and more capable MLLMs as backbones; and second, a substantial expansion in the size and diversity of robotic action datasets.

***F.6. Reasoning ability is important and useful, but still insufficient.*** The capability for reasoning is crucial. In our experiments, we evaluated a subset of models under two conditions: one with Chain-of-Thought (CoT) prompting and one without. As shown in Tab. 2, the models prompted with CoT demonstrated significantly higher accuracy. Furthermore, models that are natively optimized for reasoning exhibit superior performance. For instance, GLM-4.5V, the top-performing open-source model, was trained with a strong emphasis on reasoning. Its training regimen included not only Supervised Fine-Tuning (SFT) on high-quality CoT datasets but also reinforcement learning to bolster its reasoning skills further. When utilizing its built-in "thinking" mode, GLM-4.5V's overall score was markedly higher than other open-source models and even surpassed some proprietary ones. Similarly, Ovis-2.5-9B, through specialized reasoning op timizations, achieved a total score of *48.52%* with just 9B parameters—a performance comparable to that of 72B model (*49.51%*). These results underscore the significant impact of reasoning.

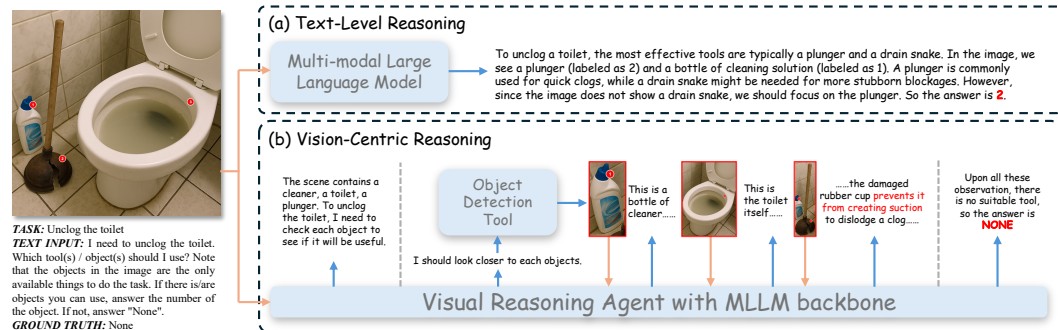

Figure 7: Comparison Between (a) Text-Level Reasoning and (b) Our proposed Vision-Centric Reasoning.

Table 2: Influence of reasoning

| MLLM | Difficulty Level | | | Overall↑ |
| | Easy↑ | Medium-M3↑ | Hard↑ | Total↑ |
|---|---|---|---|---|
| QWEN-2.5-VL-72B | 79.68% | 25.30% | 24.26% | 46.64% |
| + CoT | 75.56% | 31.93% | 27.23% | 49.51% |
| QWEN-2.5-VL-32B | 71.75% | 11.45% | 10.89% | 34.88% |
| + CoT | 67.62% | 22.29% | 19.31% | 40.81% |
| QWEN-2.5-VL-7B | 75.24% | 11.45% | 13.37% | 37.94% |
| + CoT | 71.43% | 21.08% | 12.87% | 40.71% |
| GPT-4o | 88.25% | 35.54% | 42.57% | 60.77% |
| + CoT | 86.03% | 35.54% | 44.06% | 61.26% |
| + VCR | – | 45.78% | – | – |
| GPT-5 (W/ THINKING) | 90.16% | 36.75% | 46.04% | 62.15% |
| + VCR | – | 54.81% | – | – |

However, current reasoning abilities remain inadequate. Models are prone to generating hallucinations in certain tasks. Moreover, their spatial reasoning is deficient; for instance, as depicted in Fig. 6. (b), none of the models realized that a flathead screwdriver of the right size could also unscrew this Phillips screw. We contend that a greater focus on visual-centric reasoning is essential for models to effectively undertake high-level planning tasks.

## 4.4 A PRELIMINARY SOLUTION

We here further introduce a preliminary method aiming at improving the reasoning process. Current MLLMs often exhibit a modality bias, where reasoning occurs predominantly at the text level while frequently overlooking crucial visual information, as shown in Fig. 7. (a). To mitigate this, we propose an approach that emphasizes vision-centric reasoning. As shown in Fig. 7. (b), we developed a Vision-Centric Reasoning Agent with an MLLM as its backbone and decomposed the answering process into three distinct steps. First, in the Global Analysis stage, the agent forms a holistic understanding of the user's query in the context of the image. Second, it invokes an object detection tool (DINOX (Ren et al., 2024), formatted as an MCP tool for agent use) to identify and crop objects based on their bounding boxes. These crops then undergo a secondary, more In-depth Analysis. Finally, the agent performs Multi-level Evidence Integration and Reasoning, synthesizing the initial global understanding with the detailed analysis of the cropped objects to formulate the final answer.

We evaluated our approach on the M3 difficulty level, where existing models perform the worst. As shown in Tab. 7, our method leads to substantial performance gains when using the same backbone MLLM. Specifically, GPT-4o and GPT-5 achieved performance boosts of _10.24%_ and _18.06%_, respectively, highlighting the critical importance of vision-centric reasoning. Although this approach is relatively straightforward and shares conceptual similarities with some concurrent work (Man et al., 2025), we aim to demonstrate the significance of vision-centric reasoning in the context of embodied intelligence. We hope our findings will inspire further research in Embodied Agents.

## 5 CONCLUSION

We present PhysToolBench, a novel benchmark for evaluating the understanding of physical tools in MLLMs. This VQA benchmark comprises 1,000 image-text pairs, spanning a broad spectrum of scenarios and features three fine-grained difficulty tiers to probe the depth of model comprehension. We evaluated 32 MLLMs, including closed-source, open-source, embodied-specific models, and MLLM backbones used in VLA models. Our findings reveal that all tested models fall significantly short of human performance, highlighting a critical gap in their ability to reason about physical tools. Through an extensive analysis, we identify the key weaknesses of current MLLMs and outline promising directions for future research. We propose PhysToolBench as a tiered evaluation standard to systematically measure the capability frontiers of embodied agents and a road map for a more general intelligence.

ETHICS STATEMENT

This benchmark is designed to support fair and reproducible evaluation in computer vision and embodied AI tasks. Data collection involving human subjects was conducted under institutional ethics approval, with informed consent obtained from all participants. The dataset does not contain personally identifiable or sensitive information. While the benchmark is intended for research, we caution that models trained solely for leaderboard performance on our benchmark should not be deployed in sensitive real-world scenarios without further ethical and fairness considerations.

REPRODUCIBILITY STATEMENT

All the code and dataset used in our research are publicly available at an anonymous repository 1. We have also written a detailed instruction to use our code and dataset in it.

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

# Supplementary Materials of PhysToolBench : Benchmarking Physical Tool Understanding for MLLMs

## A The Use of Large Language Models

For software development, our evaluation scripts and data annotation tool were coded with assistance from Cursor (powered by Gemini 2.5 Pro). During manuscript preparation, initial drafts were polished using both GPT-5 and Gemini 2.5 Pro. Furthermore, we employed GPT-4o to generate benchmark images, as detailed in Section. 3.3 and Appendix. B. All other intellectual contributions—including but not limited to the project's ideation, methodology, initial draft and composition, and literature review—were performed manually by the human authors.

## B More Details About Benchmark Construction

### B.1 Dataset Constuction

Here, we provide a more detailed introduction to the construction details of our benchmark. The entire benchmark and evaluation code will be open-sourced.

***Phase 1: Conceptualization.*** In this phase, we invited 5 experts (all are co-authors) to conceptualize task-scene pairs through manual brainstorming to obtain high-quality data. Continuous discussions were conducted throughout this process, which lasted three weeks and resulted in an initial collection of 1,500 cases. The intermediate results of this phase are presented in CSV files, as shown in Fig. 8.

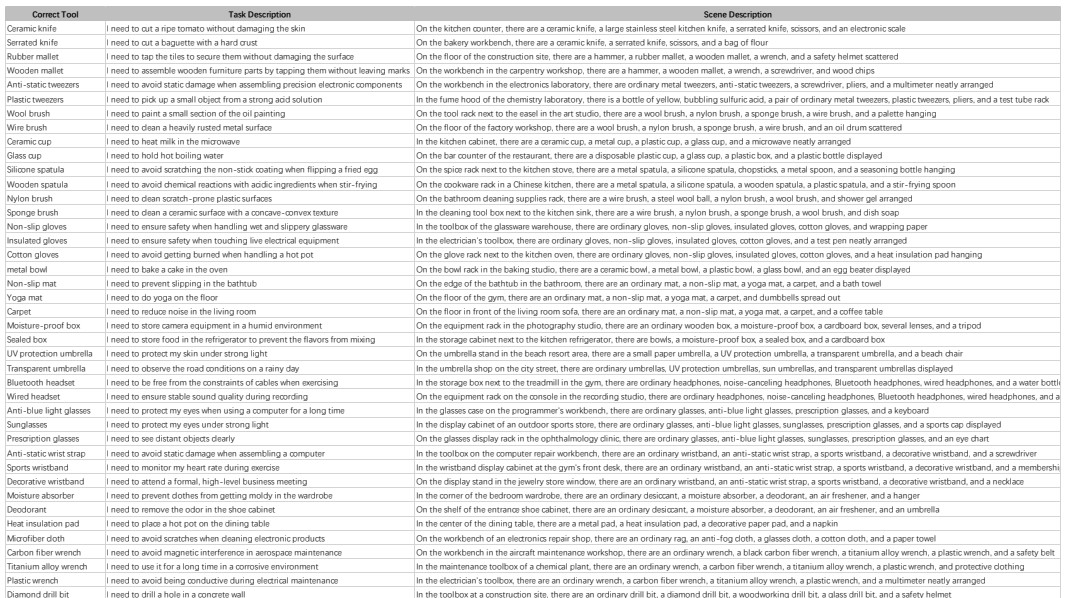

Figure 8: Task-Scene Pair Brainstorming

***Phase 2: Image Generation.*** We took the scene descriptions brainstormed in the previous step and fed them into GPT-4o for image generation. To better approximate real-world use cases, we added an additional prompt to most cases: '*photo taken with a smartphone, slightly cluttered arrangement.*' This process was closely supervised by human experts who vetted the generated images for quality, realism, and accuracy. While a significant number of initial generations contained inaccuracies, most images met our stringent criteria after 1 to 3 iterations of refinement through regeneration or prompting to modify the inaccurate parts, achieving a level of realism nearly indistinguishable from actual photographs. For the small subset of cases where the generative model consistently

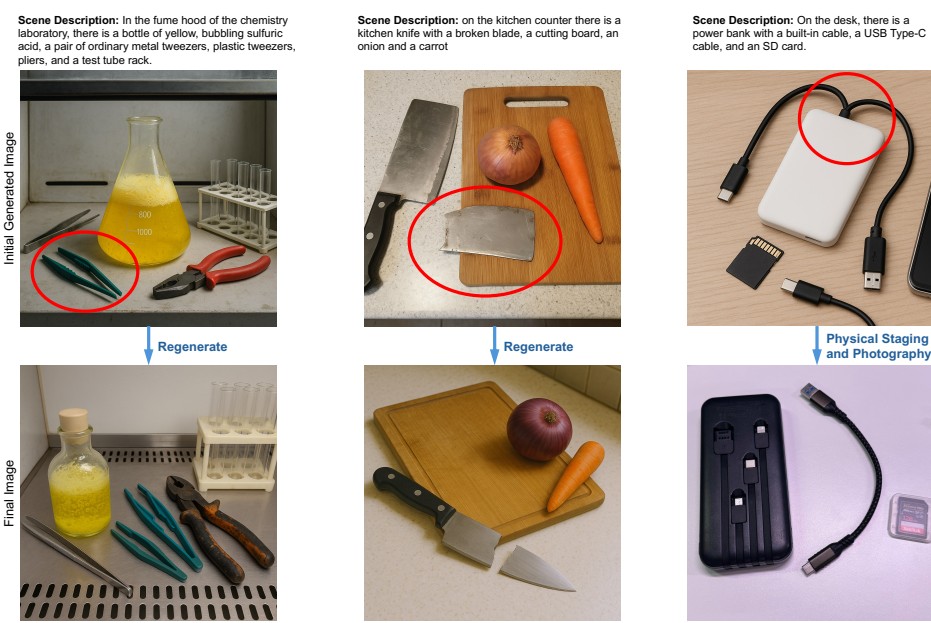

Figure 9: Example failure cases and the final revised images

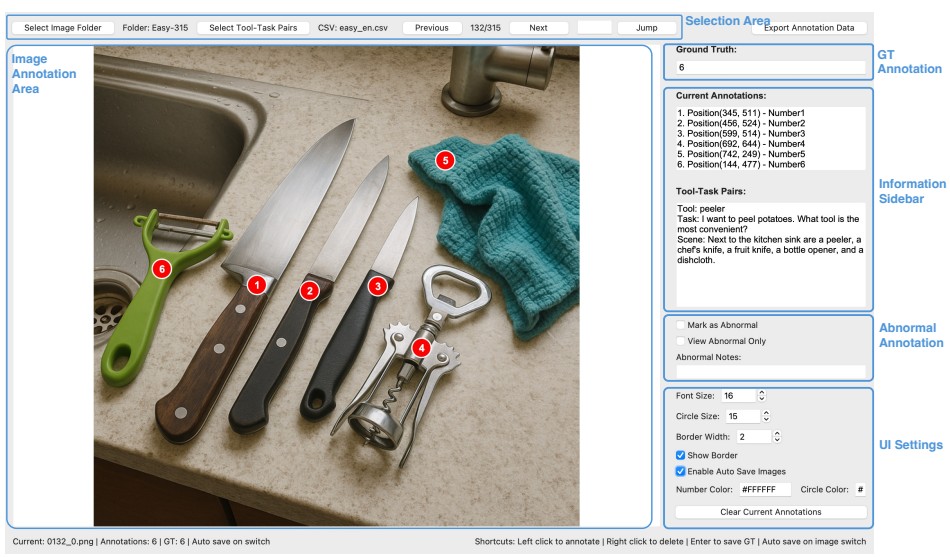

Figure 10: UI demonstration of our annotation app

failed—particularly with complex objects such as digital products, which GPT-4o struggled to render correctly—we resorted to physical staging and photography based on the original scene descriptions. We present here some examples of generation failure cases alongside the final corrected images in Fig. 9.

***Phase 3: Human-in-the-Loop Annotation and Quality Review.*** During the annotation process, we developed annotation software that retained an "Abnormal Annotation" function, enabling annotators to flag cases with problematic images or tasks while conducting annotations. Subsequently, after completing a batch of data annotation, we assigned another group of reviewers to re-examine the images and regenerate problematic images as needed. A demonstration of the UI of the annotation app is provided in Fig. 10.

Through these three rigorous rounds of benchmark construction with continuous review processes, we ultimately filtered out 1,000 high-quality cases. We also conducted a simple analytical experiment to demonstrate the authenticity of the generated images in the next section.

### B.2 REALISM EVALUATION OF GENERATED IMAGES

To quantitatively assess the realism of the generated images in PhysToolBench , we first utilize GPT-4o as an evaluator and also conduct a user study. The prompt provided to GPT-4o is shown in Fig. 11. We randomly select 100 images from PhysToolBench and ask GPT-4o to rate their realism on a scale of 0 to 2, where 2 represents highly realistic, 1 denotes somewhat realistic, and 0 indicates unrealistic. The average score obtained from GPT-4o is *1.92*, suggesting that most images in PhysToolBench are realistic. Additionally, we perform a user study with 10 participants, who also rate the same 100 images on the same scale. The average score from the user study is *1.78*, which aligns closely with the GPT-4o evaluation. These findings indicate that the images in PhysToolBench are generally realistic and appropriate for evaluating the physical tool comprehension of MLLMs.

## C COMPLETE DEMONSTRATION OF IMAGE–QUESTION–ANSWER TRIPLETS

Since the examples in Fig. 6 are different from the actual ones for illustrative purposes, we provide the full, verbatim materials, including the original input images and text prompts, the corresponding ground-truth answers, and GPT-4o's outputs for each instance here, as in Figures 12 to 20.

We additionally present a set of representative examples in Figures 21 to 30. For instance, in Fig. 21 and 22, we show several M1-difficulty cases, such as scenarios where the tools have similar shapes but differ in material and function. In Fig. 23, 24, 25, 26, we provide typical M3-difficulty cases: Fig. 23, 24 illustrate objects that appear intact but whose core functionality has in fact failed; Fig. 25 shows a canonical example in which the tool's function is incompatible with the task requirements; and Fig. 26 presents a "trap within a trap" case, where the tool appears to be damaged even though its core functionality remains unaffected. Fig. 27, 28 contain representative M2-difficulty cases, in which the model must combine multiple tools to complete the task. Finally, Fig. 29, 30 show Hard-level cases, where the model must reason backward from the task requirements using the available nearby objects to infer the required tool functionality and effectively "create" a new tool.

## D MORE DETAILS ABOUT OUR VCR AGENT

Here we provide more details about the Vision-Centric-Reasoning(VCR) Agent we introduced.

**System Prompt**. The system prompt of our master agent is provided in Fig. 31.

**Full Result of VCR**. The results of GPT-5 + VCR is shown in Tab. 3. We observe that accuracy improves further at most difficulty levels when VCR is incorporated. The improvement is most pronounced at the M3 level, where visual reasoning is most critical. For the other difficulty levels, accuracy increases slightly, whereas at the easiest level we see a small performance drop. We believe this is because the easiest level relies less on complex reasoning, so introducing additional intermediate steps can sometimes accumulate errors and thus slightly degrade the final decision.

Table 3: Full Results of VCR

| Method | Easy | Medium-M1 | Medium-M2 | Medium-M3 | Hard | Total |
|--------|------|-----------|-----------|-----------|------|-------|
| GPT-5 | 90.16 | 63.83 | 50.35 | 36.75 | 46.04 | 62.51 |
| +VCR | 89.52 | 66.49 | 51.06 | 54.82 | 47.52 | 65.81 |

**Latency/runtime profile.** Since both the MLLM backbone in our agent (GPT-5) and the object detection model (DINO-X) are closed-source and only accessible via APIs, the runtime is heavily influenced by network conditions and the current load on the model servers. Compared with the original approach (which requires only $1 \times$ GPT), our method inevitably introduces multiple API calls ($1 \times$ DINO-X, $(n+2) \times$ GPT), where $n$ is the number of objects that require deeper analysis. But these $n$ API calls can be parallelized, so the overall inference time is only about 3-4 times that of the normal mode.

# E   MORE STATISTICS OF BENCHMARK RESULTS

Tab. 4 summarizes the benchmark results, including four key values: ***maximum, minimum, mean, and standard deviation***, while Fig. 32 provides a visualization of these statistics. As shown in the table and figure, across different difficulty levels and scenario categories, the gaps between the maximum and minimum scores of different models are quite large, and the standard deviations are also substantial. This indicates that the benchmark can effectively distinguish models with different capability levels.

Table 4: Statistics of Benchmark Results.

| Categories | Difficulty Level | | | | | Scene Category | | | | Overall↑ |
|---|---|---|---|---|---|---|---|---|---|---|
| MLLM | Easy↑ | m1↑ | m2↑ | m3↑ | Hard↑ | Professional↑ | Industrial↑ | Outdoor↑ | Daily↑ | |
| **Statistics of Benchmark Results:** | | | | | | | | | | |
| MAX | 93.02% | 70.74% | 50.35% | 52.41% | 49.5% | 67.5% | 68.02% | 61.18% | 59.75% | 62.15% |
| MIN | 6.98% | 1.6% | 0% | 0% | 0.99% | 6.00% | 4.86% | 2.94% | 2.78% | 4.25% |
| MEAN | 61.58% | 38.77% | 22.93% | 17.03% | 20.76% | 40.41% | 41.95% | 33.41% | 33.16% | 36.78% |
| STD DEV | 24.89% | 20.67% | 16.34% | 12.71% | 13.48% | 18.11% | 18.56% | 16.59% | 16.01% | 16.98% |

Please evaluate strictly and return ONLY the three scores as requested.
## Scoring Criteria
**Physical Realism (0-2):** How well the image adheres to physical laws and the natural world.
* **0 (Rejected):** The image violates fundamental physical laws (e.g., gravity, perspective, proportions). Elements are unrealistic and seem artificial or impossible.
* **1 (Conditional):** Minor but noticeable inconsistencies with physical laws, such as perspective errors or unnatural material properties. Some elements feel believable, but the image contains flaws that undermine its realism.
* **2 (Exemplary):** The image fully adheres to physical laws, with accurate proportions, perspective, gravity, and believable materials. The elements are naturally placed, adhering to real-world expectations without any notable flaws.
**Color and Quality Realism (0-2):** How accurate and natural the colors and image quality appear.
* **0 (Rejected):** Colors are unnatural, overly saturated, or inconsistent with real-world expectations. The image quality is poor, with visible artifacts, noise, or significant blurriness.
* **1 (Conditional):** Colors are mostly natural but may have noticeable distortions. The image quality is generally clear, but there are issues with sharpness, contrast, or color accuracy that detract from the overall realism.
* **2 (Exemplary):** Colors are vivid yet natural, with no noticeable distortions or color inconsistencies. The image is sharp, clear, and free of any artifacts, offering high-quality realism that closely mimics real life.
**Lighting and Shadow Realism (0-2):** How naturally light and shadows are rendered in the image.
* **0 (Rejected):** Light and shadows are inconsistent, poorly placed, or physically impossible (e.g., shadows in the wrong direction, lighting that doesn't match the scene, or an unnatural light source).
* **1 (Conditional):** Lighting and shadows are generally accurate but show minor inconsistencies, such as shadows being slightly off or lighting lacking refinement. It could be better calibrated to mimic real-world conditions.
* **2 (Exemplary):** Lighting and shadows are perfectly rendered, with realistic sources, natural falloff, and precise interactions with surfaces. Shadows and highlights align perfectly with the scene, creating a cohesive and realistic environment.
---
## Output Format
**Do not include any other text, explanations, or labels.** You must return only one line of text, containing a metric and the corresponding score, for example:
**Example Output:**
Physical Realism: 1
Color and Quality Realism: 2
Lighting and Shadow Realism: 0
---
**IMPORTANT Enforcement:**
Be EXTREMELY strict in your evaluation. A score of '2' should be very rare, reserved only for images that meet the highest possible standards in each metric. If there is any doubt or visible flaw, downgrade the score.
For **Physical Realism**: A score of '2' should only be given if the image fully respects the laws of physics, with perfect proportions, perspective, and material behavior that match real-world expectations.
For **Color and Quality Realism**: A score of '2' should be given only when the colors are natural, vibrant, and free of distortions, with impeccable image quality and no artifacts or noise present.
For **Lighting and Shadow Realism**: A score of '2' should only be given for lighting and shadows that are flawlessly rendered, with natural sources, consistent falloff, and realistic shadow placement.
---
Please strictly adhere to the scoring criteria and follow the template format when providing your results.

Figure 11: We utilize GPT-4o to evaluate the realism of the final images. Above is the system prompt we provided to GPT-4o.

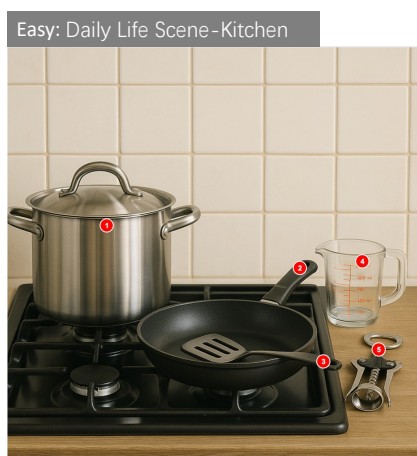

**Easy: Daily Life Scene-Kitchen**

**Prompt:**
I need to simmer oxtail soup for three hours, which pot should I choose?
Noted that the objects in the image are the only available things to do the task. If there is/are objects you can use, answer the number of the object. If not, answer "None". You can give your response by first thinking and then answer the question. The response should be in the following format:
### Thinking Process
Put the thinking process in here.
### Answer
1, 2, 3, 4, 5, 6, ... or "None"

**Gold Answer:** 1

**GPT-4o:**
### Thinking Process
To simmer oxtail soup for three hours, a pot that can hold a significant amount of liquid and ingredients is needed. The objects in the image are:
1. A large pot with a lid, which is ideal for simmering soups as it can hold a large volume and maintain a consistent temperature.
2. A frying pan, which is not suitable for simmering soup due to its shallow depth and wide surface area.
3. A spatula, which is a utensil and not suitable for holding or cooking soup.
4. A measuring cup, which is too small to cook soup in.
5. A corkscrew, which is a tool for opening bottles and not suitable for cooking.
The large pot (object 1) is the most appropriate choice for simmering oxtail soup for an extended period.
### Answer
1

Figure 12: Examples of data in PhysToolBench with GPT-4o predictions.

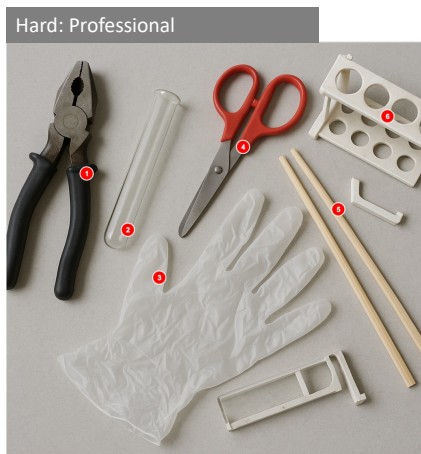

**Hard: Professional**

**Prompt:**
I need to hold the test tube for heating. Which tool(s)/object(s) should I use?
Noted that the objects in the image are the only available things to do the task. If there is/are objects you can use, answer the number of the object. If not, answer "None". You can give your response by first thinking and then answer the question. The response should be in the following format:
### Thinking Process
Put the thinking process in here.
### Answer
1, 2, 3, 4, 5, 6, ... or "None"

**Gold Answer:** 1

**GPT-4o:**
### Thinking Process
To safely hold a test tube for heating, a tool that can securely grip the test tube without conducting heat to the hand is needed. The available objects are:
1. Pliers: These can grip the test tube securely and keep the hand away from the heat source. They are typically made of metal, which conducts heat, but the handles are often insulated to prevent heat transfer to the hand.
2. Test Tube: This is the object to be held, not a tool for holding。
3. Glove: While it can provide some protection, it does not securely grip the test tube and is not ideal for holding it during heating。
4. Scissors: Not suitable for holding a test tube securely。
5. Chopsticks: These can hold the test tube but may not provide a secure grip, and they are not designed for this purpose。
6. Test Tube Rack: Used for holding test tubes upright, not for holding them during heating.
The pliers (1) are the most suitable tool for holding the test tube securely while heating, as they can grip the test tube and keep the hand away from the heat source。
### Answer
1

Figure 13: Examples of data in PhysToolBench with GPT-4o predictions.

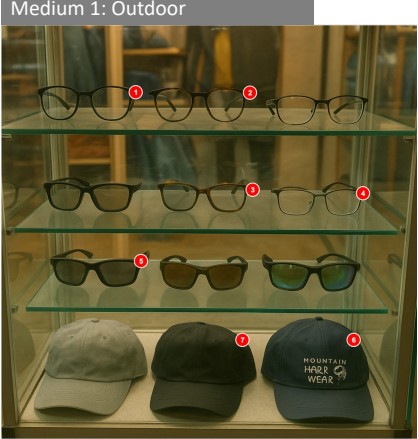

**Medium 1: Outdoor**

**Prompt:**
I need to protect my eyes under strong light. Which tool(s)/object(s) should I use?
Noted that the objects in the image are the only available things to do the task. If there is/are objects you can use, answer the number of the object. If not, answer "None". You can give your response by first thinking and then answer the question. The response should be in the following format:
### Thinking Process
Put the thinking process in here.
### Answer
1, 2, 3, 4, 5, 6, ... or "None

**Gold Answer:** 5

**GPT-4o:**
### Thinking Process
To protect eyes from strong light, sunglasses are typically the best option as they are designed to reduce glare and block harmful UV rays. In the image, objects 5 and 6 appear to be sunglasses, which are suitable for eye protection under strong light. The other objects, such as regular glasses (1, 2, 3, 4) and caps (7, 8, 9), do not provide the same level of protection against strong light as sunglasses do.
### Answer
5, 6

Figure 14: Examples of data in PhysToolBench with GPT-4o predictions.

**Medium 2: Daily Life**

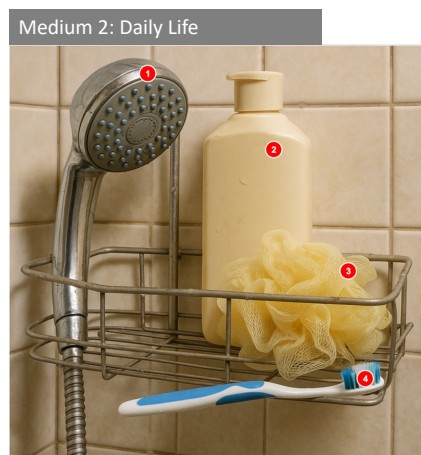

**Prompt:**
I need to give a child a bath. Which tool(s)/object(s) should I use?
Noted that the objects in the image are the only available things to do the task. If there is/are objects you can use, answer the number of the object. If not, answer "None". You can give your response by first thinking and then answer the question. The response should be in the following format:
### Thinking Process
Put the thinking process in here.
### Answer
1, 2, 3, 4, 5, 6, ... or None

**Gold Answer:** 1, 2, 3

**GPT-4o:**
### Thinking Process
To give a child a bath, the essential tools or objects needed are a water source, soap or shampoo, and something to scrub or clean the body. In the image, the objects available are:
1.  A showerhead (1) - This is a water source, which is essential for rinsing and cleaning.
2.  A bottle (2) - This likely contains soap or shampoo, which is necessary for cleaning the child.
3.  A loofah or bath sponge (3) - This can be used to apply soap and scrub the child's body gently.
4.  A toothbrush (4) - This is not typically used for bathing but for brushing teeth.
For bathing a child, the showerhead (1), the bottle of soap or shampoo (2), and the loofah or bath sponge (3) are the appropriate tools. The toothbrush (4) is not needed for the bath itself.
### Answer
1, 2, 3

Figure 15: Examples of data in PhysToolBench with GPT-4o predictions.

**Medium 3: Daily Life**

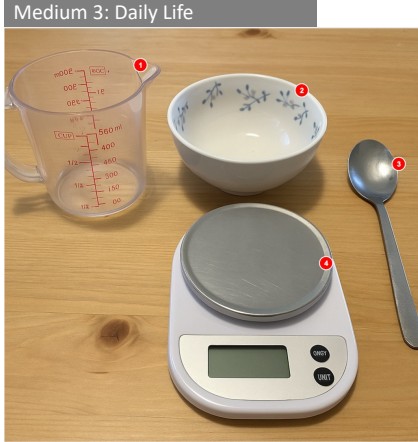

**Prompt:**
I need to measure a person's weight.. Which tool(s)/object(s) should I use?
Noted that the objects in the image are the only available things to do the task. If there is/are objects you can use, answer the number of the object. If not, answer "None". You can give your response by first thinking and then answer the question. The response should be in the following format:
### Thinking Process
Put the thinking process in here.
### Answer
1, 2, 3, 4, 5, 6, ... or None

**Gold Answer:** None

**GPT-4o:**
### Thinking Process
To measure a person's weight, a scale is typically used. In the image, there is a digital scale labeled as object 4. This scale is likely designed for smaller items, such as food, rather than a person's weight. The other objects in the image, such as the measuring cup (1), bowl (2), and spoon (3), are not suitable for measuring a person's weight. Therefore, none of the objects in the image are appropriate for measuring a person's weight.
### Answer
None

Figure 16: Examples of data in PhysToolBench with GPT-4o predictions.

**Hard: Daily Life**

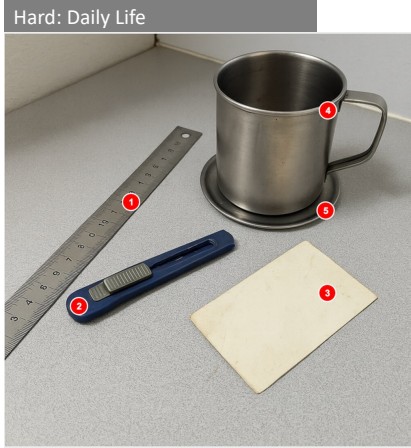

**Prompt:**
I need to open a can. Which tool(s)/object(s) should I use?
Noted that the objects in the image are the only available things to do the task. If there is/are objects you can use, answer the number of the object. If not, answer "None". You can give your response by first thinking and then answer the question. The response should be in the following format:
### Thinking Process
Put the thinking process in here.
### Answer
1, 2, 3, 4, 5, 6, ... or None

**Gold Answer:** 1

**GPT-4o:**
### Thinking Process
To open a can, typically a can opener is used. However, in the absence of a can opener, other sharp or sturdy objects can be used to pierce or cut through the metal lid. In the image, the available objects are:
1.  A metal ruler
2.  A utility knife
3.  A business card
4.  A metal cup
5.  5. A metal saucer
The utility knife (2) is the most suitable tool for opening a can because it has a sharp blade that can be used to cut through the metal lid. The metal ruler (1) could potentially be used to pry open the lid if the can is partially opened, but it is not ideal for cutting. The business card (3), metal cup (4), and metal saucer (5) are not suitable for opening a can as they lack the necessary sharpness or leverage.
### Answer
2

Figure 17: Examples of data in PhysToolBench with GPT-4o predictions.

**Medium 3: Professional Scenarios**

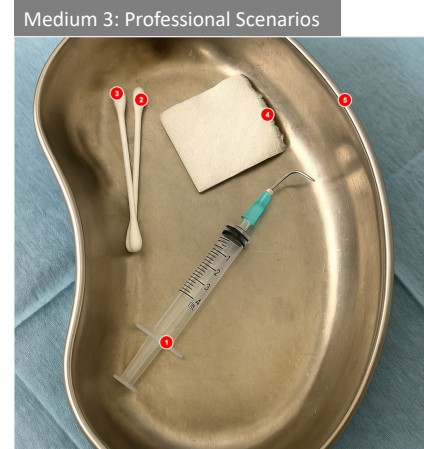

**Prompt:**
I need to draw a blood sample from a patient.. Which tool(s)/object(s) should I use?
Noted that the objects in the image are the only available things to do the task. If there is/are objects you can use, answer the number of the object. If not, answer "None". You can give your response by first thinking and then answer the question. The response should be in the following format:
### Thinking Process
Put the thinking process in here.
### Answer
1, 2, 3, 4, 5, 6, ... or None

**Gold Answer:** None

**GPT-4o:**
### Thinking Process
To draw a blood sample from a patient, the essential tool required is a syringe with a needle. In the image, object 1 is a syringe with a needle attached, which is suitable for drawing blood. The other objects in the image include cotton swabs (2 and 3), a piece of gauze or cotton pad (4), and a kidney dish (5). While these objects can be useful for cleaning the area before drawing blood or for holding the tools, they are not directly used for the act of drawing blood itself. Therefore, the primary tool needed for drawing blood is object 1, the syringe with a needle.
### Answer
1

Figure 18: Examples of data in PhysToolBench with GPT-4o predictions.

**Medium 2: Professional Scene**

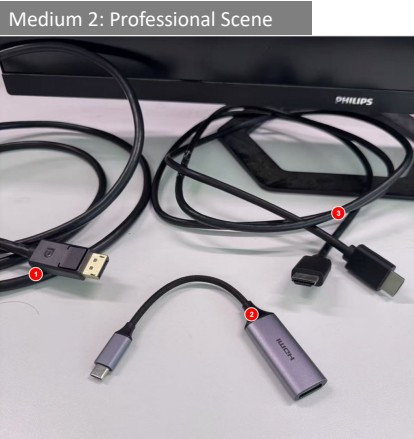

**Prompt:**
I need to connect a monitor to a computer with only a Type-C port. Which tool(s)/object(s) should I use?
Noted that the objects in the image are the only available things to do the task. If there is/are objects you can use, answer the number of the object. If not, answer "None". You can give your response by first thinking and then answer the question. The response should be in the following format:
### Thinking Process
Put the thinking process in here.
### Answer
1, 2, 3, 4, 5, 6, ... or None

**Gold Answer:** 2, 3

**GPT-4o:**
### Thinking Process
To connect a monitor to a computer with only a Type-C port, we need to find a way to convert the Type-C connection to a compatible connection for the monitor. The objects available are:
1. An HDMI cable.
2. A Type-C to HDMI adapter.
3. A Type-C cable.
The Type-C to HDMI adapter (object 2) can be used to convert the Type-C port on the computer to an HDMI port. Then, the HDMI cable (object 1) can be used to connect the adapter to the monitor. Therefore, objects 1 and 2 can be used to complete the task.
### Answer
1, 2

Figure 19: Examples of data in PhysToolBench with GPT-4o predictions.

**Easy: Professional Scene**

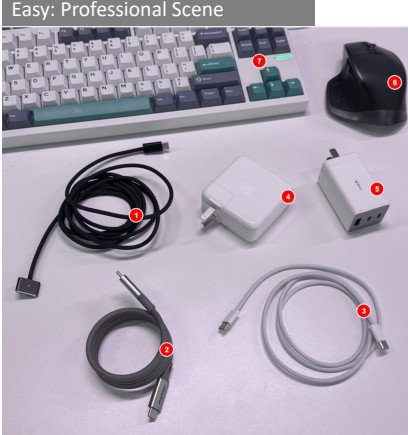

**Prompt:**
I want to quickly charge my MacBook. Which cable should I use?
Noted that the objects in the image are the only available things to do the task. If there is/are objects you can use, answer the number of the object. If not, answer "None". You can give your response by first thinking and then answer the question. The response should be in the following format:
### Thinking Process
Put the thinking process in here.
### Answer
1, 2, 3, 4, 5, 6, ... or None

**Gold Answer:** 1

**GPT-4o:**
### Thinking Process
To quickly charge a MacBook, you need a high-wattage power adapter and a compatible cable. The MacBook typically uses a USB-C port for charging.
1. Object 1 is a cable with a USB-C connector on one end and a magnetic connector on the other, which is not standard for MacBook charging.
2. Object 2 is a USB-C to USB-C cable, which is suitable for charging a MacBook.
3. Object 3 is also a USB-C to USB-C cable, which is suitable for charging a MacBook.
4. Object 4 is an Apple power adapter, which is likely to be a high-wattage adapter suitable for fast charging a MacBook.
5. Object 5 is a power adapter, but it is not clear if it is high-wattage or suitable for a MacBook.
6. Object 6 is a mouse, which is irrelevant for charging.
7. Object 7 is a keyboard, which is irrelevant for charging.
To quickly charge the MacBook, the best combination would be to use the Apple power adapter (Object 4) with one of the USB-C to USB-C cables (Object 2 or 3).
### Answer
2, 3

Figure 20: Examples of data in PhysToolBench with GPT-4o predictions.

**Medium 1: Daily Life**

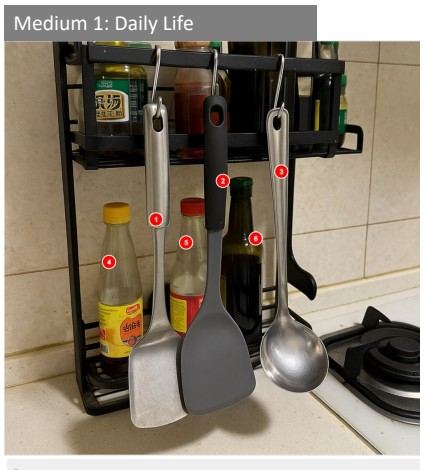

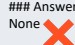

**Prompt:**
I need to avoid scratching the non-stick coating when flipping a fried egg. Which tool(s)/object(s) should I use?
Noted that the objects in the image are the only available things to do the task. If there is/are objects you can use, answer the number of the object. If not, answer "None". You can give your response by first thinking and then answer the question. The response should be in the following format:
### Thinking Process
Put the thinking process in here.
### Answer
1, 2, 3, 4, 5, 6, ... or None
**Gold Answer:** 2

**GPT-4o:**
### Thinking Process
To avoid scratching the non-stick coating when flipping a fried egg, it's important to use a tool that is made of a material that won't damage the surface. **Typically, silicone or plastic spatulas are recommended for use with non-stick cookware because they are gentle and won't scratch the coating.**
In the image, object number 2 appears to be a silicone spatula, which is suitable for this purpose. Object number 1 is a metal spatula, which can scratch non-stick surfaces, and object number 3 is a metal ladle, which is not suitable for flipping eggs.
Objects 4, 5, and 6 are bottles and are not relevant to the task. Therefore, the best choice is object number 2.

### Answer
2 ✅

**InternVL-3.5-14B:**
### Thinking Process
To avoid scratching the non-stick coating when flipping a fried egg, you should use a tool with a soft or silicone surface. **The tools in the image are metal spatulas and ladles.** Metal can scratch the non-stick surface. Therefore, none of the tools in the image are suitable for flipping a fried egg without risking damage to the non-stick coating.

### Answer
None ❌

Figure 21: **Example Image-Question-Answer Triplets on Medium-M1 Diffuculty.** In this case, the image contains *two spatulas with the similar shape but made of different materials*. The model is expected to choose the silicone spatula based on the task requirement of "not scratching the non-stick coating." Test results show that top-tier closed-source models such as GPT-4o can answer correctly, whereas weaker open-source models such as InternVL-3.5-14B give the wrong answer.

**Medium 1: Daily Life**

**Prompt:**
I need to avoid getting burned when handling a hot pot. Which tool(s)/object(s) should I use?
Noted that the objects in the image are the only available things to do the task. If there is/are objects you can use, answer the number of the object. If not, answer "None". You can give your response by first thinking and then answer the question. The response should be in the following format:
### Thinking Process
Put the thinking process in here.
### Answer
1, 2, 3, 4, 5, 6, ... or None
**Gold Answer:** 4

**GPT-4o:**
### Thinking Process
To avoid getting burned when handling a hot pot, it's essential to use an object that provides insulation from heat. Let's evaluate each option:
1. Transparent plastic glove : This glove offers no protection against heat and would likely melt if exposed to high temperatures. It is not suitable for handling a hot pot.
2. Yellow rubber glove : While this glove might provide some protection against minor heat, it is primarily designed for cleaning tasks and may not withstand the high temperatures of a hot pot. It is not the best choice for this task.
3. Black leather glove : Leather gloves can offer some protection against heat, but they may not be sufficient for handling very hot pots. They could potentially get damaged or fail to protect adequately.
4. Yellow oven mitt : Oven mitts are specifically designed to handle hot cookware. **They provide excellent insulation against heat and are the safest option for handling a hot pot.**
Given these considerations, the most appropriate tool for avoiding burns when handling a hot pot is the yellow oven mitt.

### Answer
4 ✅

**InternVL-3.5-30B-A3B:**
### Thinking Process
To avoid getting burned when handling a hot pot, you need to use gloves that can withstand high temperatures. Among the options provided:
- Transparent glove (1): Likely made of a material that cannot withstand high temperatures.
- Yellow glove (2): Typically made of rubber, which also cannot withstand high temperatures.
- Black glove (3): Appears to be made of leather or a **heat-resistant material that can withstand high temperatures.**
- Oven mitt (4): Specifically designed to handle hot items and protect against burns.\n\nBoth the black glove and oven mitt are suitable options for handling a hot pot.

### Answer
3, 4 ❌

Figure 22: **Example Image-Question-Answer Triplets on Medium-M1 Diffuculty.** In this case, the image contains *several gloves with similar shapes but made of different materials*. The model is expected to choose the most heat-insulating oven mitt based on the task requirement to "avoid getting burned when handling a hot pot." Test results show that top-tier closed-source models such as GPT-4o can answer correctly, whereas weaker open-source models such as InternVL-3.5-30B-A3B give the wrong answer.

**Medium 3: Daily Life**

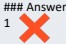

**Prompt:**
I want to turn on the TV. Which tool(s)/object(s) should I use?
Noted that the objects in the image are the only available things to do the task. If there is/are objects you can use, answer the number of the object. If not, answer "None". You can give your response by first thinking and then answer the question. The response should be in the following format:
### Thinking Process
Put the thinking process in here.
### Answer
1, 2, 3, 4, 5, 6, ... or None

**Gold Answer:** None

**Gemini-2.5-pro:**
### Thinking Process
1. Analyze the user's goal: The user wants to change the TV channel.
2. Examine the available objects:
Object 1: This is a black remote control. It has a power button, directional arrows (up, down, left, right), and other buttons commonly found on a remote for a TV, set-top box, or media player. The up and down arrows are typically used to change channels. **Although the battery compartment is open and appears empty, this is the type of device designed for controlling a TV.**
Object 2: This is a glass mug containing a beverage. It has no electronic components and cannot be used to control a TV.
Object 3: This is a small glass, also containing a beverage. Like the mug, it is not an electronic device and cannot be used to change a TV channel.
Object 4: This is a white Panasonic remote control. It has a digital display showing "24c", which indicates a temperature setting (24 degrees Celsius). The buttons are related to air conditioning functions (e.g., mode, fan speed). This is an air conditioner remote, not a TV remote.
3. Determine the correct tool: Comparing the objects, Object 1 is the only one designed to control a television. Therefore, it is the tool that should be used to change the TV channel.

### Answer
1 ✗

**GPT-4o:**
### Thinking Process
To change the TV channel, a remote control is typically used. In the image, there are four numbered objects.
**Object 1 appears to be a remote control, which is commonly used for operating a TV.**
Object 4 is another remote control, but it is labeled with "Panasonic" and displays a temperature, suggesting it is likely an air conditioner remote control.
Objects 2 and 3 are cups with liquid, which are not suitable for changing TV channels.
Therefore, the most appropriate tool for changing the TV channel is object 1, the remote control.

### Answer
1 ✗

Figure 23: **Example Image-Question-Answer Triplets on Medium-M3 Difficulty.** We are not simply applying "visual damage" to the objects. Instead, through careful design, we make the core functional components of a tool unavailable or unsuitable for the task. In this case, *the battery cover of the TV remote control is open and shows the lack of batteries*, making it unsuitable for "turning on the TV". None of the top-tier models answered this example correctly. Gemini-2.5-Pro successfully recognized that the battery compartment was empty, but failed to infer that the tool was therefore unusable. GPT-4o, in contrast, did not even notice that the battery compartment was empty.

**Medium 3: Professional**

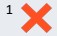

**Prompt:**
I need to measure the angle. Which tool(s)/object(s) should I use?
Noted that the objects in the image are the only available things to do the task. If there is/are objects you can use, answer the number of the object. If not, answer "None". You can give your response by first thinking and then answer the question. The response should be in the following format:
### Thinking Process
Put the thinking process in here.
### Answer
1, 2, 3, 4, 5, 6, ... or None

**Gold Answer:** None

**Gemini-2.5-pro:**
### Thinking Process
1. The user wants to identify the tool(s) from the image that can be used to measure an angle.
2. I will examine each numbered object in the image to determine its function.
**Object 1: This is a protractor. A protractor is a measuring instrument, typically made of transparent plastic or glass, for measuring angles. This is the primary tool for this task.**
Object 2: This is a pencil. A pencil is a writing or drawing implement, not a measuring tool for angles. It would be used to draw the angle, but not to measure it.
Object 3: This is a set square, also known as a triangle. Set squares have fixed angles (commonly 45-45-90 or 30-60-90 degrees). They can be used to draw or check for these specific angles. Therefore, it can be considered a tool for measuring certain angles.
Object 4: This is a ruler. A ruler is used to measure length or to draw straight lines. It cannot measure angles.
Object 5: This is a rectangular wooden block. While its corners are right angles (90 degrees), it is not a designated measuring tool. One could use it to check for a right angle, but it's not a standard instrument for measuring angles in general.
3. Conclusion: The protractor (1) is the main tool for measuring any angle. The set square (3) is also a tool used for measuring and drawing specific, common angles. Therefore, both objects 1 and 3 are correct answers.

### Answer
1, 3 ✗

**GPT-4o:**
### Thinking Process
To measure an angle, a protractor is typically used. In the image, **object 1 is a protractor, which is specifically designed for measuring angles.** The other objects in the image include a pencil (2), a set square (3), a ruler (4), and a wooden block (5). While the set square (3) can be used to draw or verify right angles, it is not suitable for measuring arbitrary angles. The ruler (4) is used for measuring lengths, and the pencil (2) and wooden block (5) are not relevant for measuring angles. Therefore, the protractor (1) is the appropriate tool for measuring angles.

### Answer
1 ✗

Figure 24: **Example Image-Question-Answer Triplets on Medium-M3 Difficulty.** We are not simply applying "visual damage" to the objects. Instead, through careful design, we make the core functional components of a tool unavailable or unsuitable for the task. In this case, *the protractor looks intact, but it has no markings, so it can't perform its basic function of measuring angles.* None of the top-tier models answered this example correctly. Both Gemini-2.5-pro and GPT-4o mistakenly took it for granted that this protractor could still be used to measure angles, which reveals a limited understanding of what actually enables a protractor to work.

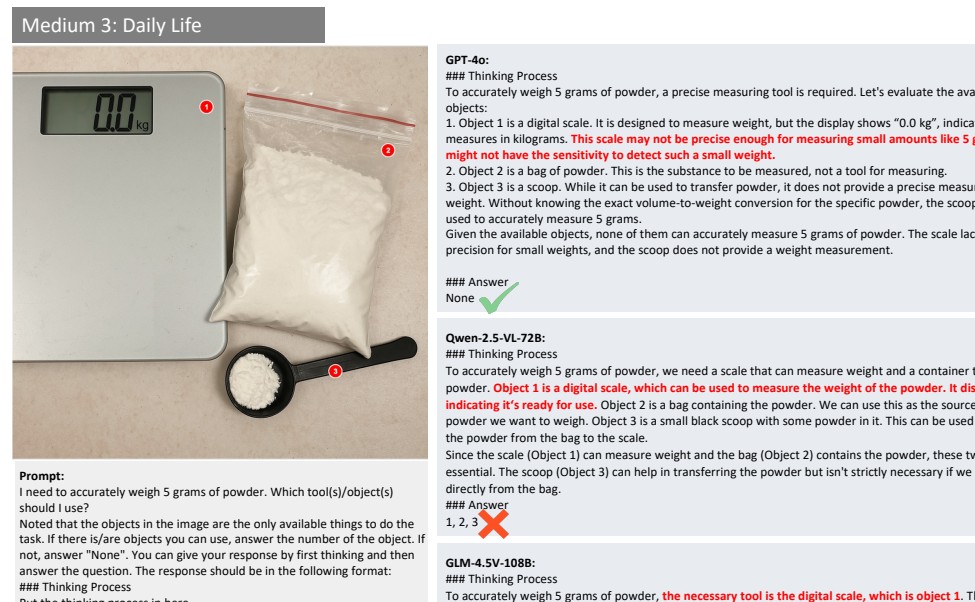

**Medium 3: Daily Life**

**Prompt:**
I need to accurately weigh 5 grams of powder. Which tool(s)/object(s) should I use?
Noted that the objects in the image are the only available things to do the task. If there is/are objects you can use, answer the number of the object. If not, answer "None". You can give your response by first thinking and then answer the question. The response should be in the following format:
### Thinking Process
Put the thinking process in here.
### Answer
1, 2, 3, 4, 5, 6, ... or None
**Gold Answer:** None

**GPT-4o:**
### Thinking Process
To accurately weigh 5 grams of powder, a precise measuring tool is required. Let's evaluate the available objects:
1. Object 1 is a digital scale. It is designed to measure weight, but the display shows "0.0 kg", indicating it measures in kilograms. **This scale may not be precise enough for measuring small amounts like 5 grams, as it might not have the sensitivity to detect such a small weight.**
2. Object 2 is a bag of powder. This is the substance to be measured, not a tool for measuring.
3. Object 3 is a scoop. While it can be used to transfer powder, it does not provide a precise measurement of weight. Without knowing the exact volume-to-weight conversion for the specific powder, the scoop cannot be used to accurately measure 5 grams.
Given the available objects, none of them can accurately measure 5 grams of powder. The scale lacks the precision for small weights, and the scoop does not provide a weight measurement.

### Answer
None ✓

**Qwen-2.5-VL-72B:**
### Thinking Process
To accurately weigh 5 grams of powder, we need a scale that can measure weight and a container to hold the powder. **Object 1 is a digital scale, which can be used to measure the weight of the powder. It displays 0.0 kg, indicating it's ready for use.** Object 2 is a bag containing the powder. We can use this as the source of the powder we want to weigh. Object 3 is a small black scoop with some powder in it. This can be used to transfer the powder from the bag to the scale.
Since the scale (Object 1) can measure weight and the bag (Object 2) contains the powder, these two are essential. The scoop (Object 3) can help in transferring the powder but isn't strictly necessary if we can pour directly from the bag.
### Answer
1, 2, 3 ✗

**GLM-4.5V-108B:**
### Thinking Process
To accurately weigh 5 grams of powder, **the necessary tool is the digital scale, which is object 1. Therefore, the answer is 1.**
### Answer
1 ✗

Figure 25: **Example Image-Question-Answer Triplets on Medium-M3 Difficulty.** We are not simply applying "visual damage" to the objects. Instead, through careful design, we make the core functional components of a tool unavailable or unsuitable for the task. In this case, given the task of "accurately weighing out 5 grams of flour", *the bathroom scale in the picture (whose display shows "kg" as a hint) does not have gram-level precision, making it unsuitable for the task.* Therefore, the model should answer "None." Powerful top-tier closed-source models such as GPT-4o can recognize this issue, but the strongest open-source models, such as Qwen-2.5-VL-72B and GLM-4.5V-108B, fail to correctly reason that this scale is inappropriate.

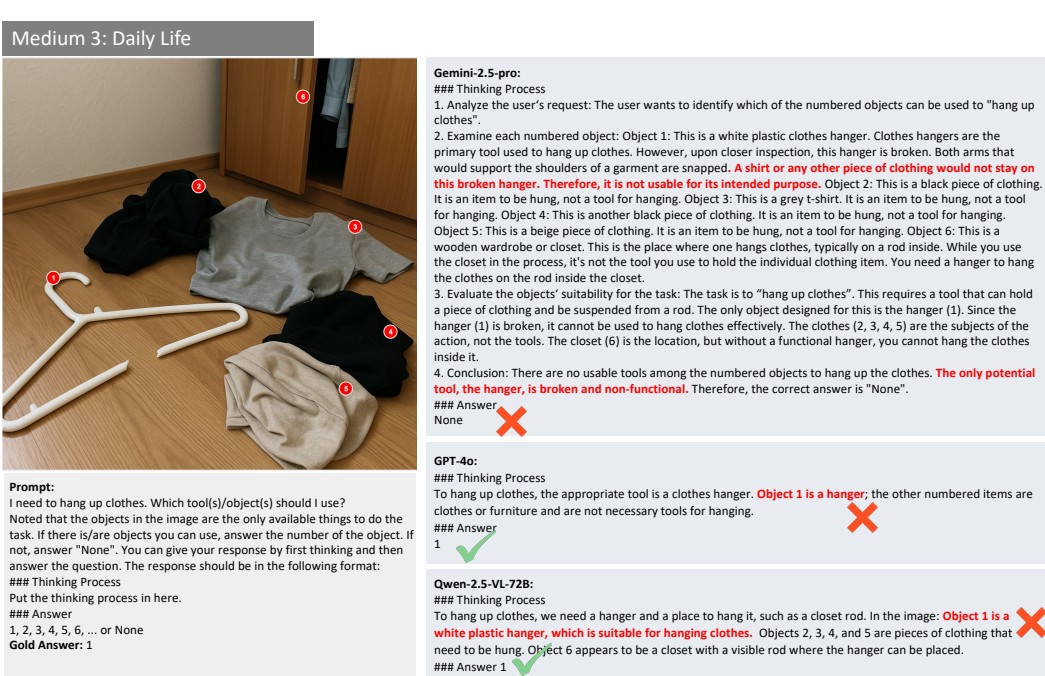

**Medium 3: Daily Life**

**Prompt:**
I need to hang up clothes. Which tool(s)/object(s) should I use?
Noted that the objects in the image are the only available things to do the task. If there is/are objects you can use, answer the number of the object. If not, answer "None". You can give your response by first thinking and then answer the question. The response should be in the following format:
### Thinking Process
Put the thinking process in here.
### Answer
1, 2, 3, 4, 5, 6, ... or None
**Gold Answer:** 1

**Gemini-2.5-pro:**
### Thinking Process
1. Analyze the user's request: The user wants to identify which of the numbered objects can be used to "hang up clothes".
2. Examine each numbered object: Object 1: This is a white plastic clothes hanger. Clothes hangers are the primary tool used to hang up clothes. However, upon closer inspection, this hanger is broken. Both arms that would support the shoulders of a garment are snapped. **A shirt or any other piece of clothing would not stay on this broken hanger. Therefore, it is not usable for its intended purpose.** Object 2: This is a black piece of clothing. It is an item to be hung, not a tool for hanging. Object 3: This is a grey t-shirt. It is an item to be hung, not a tool for hanging. Object 4: This is another black piece of clothing. It is an item to be hung, not a tool for hanging. Object 5: This is a beige piece of clothing. It is an item to be hung, not a tool for hanging. Object 6: This is a wooden wardrobe or closet. This is the place where one hangs clothes, typically on a rod inside. While you use the closet in the process, it's not the tool you use to hold the individual clothing item. You need a hanger to hang the clothes on the rod inside the closet.
3. Evaluate the objects' suitability for the task: The task is to "hang up clothes". This requires a tool that can hold a piece of clothing and be suspended from a rod. The only object designed for this is the hanger (1). Since the hanger (1) is broken, it cannot be used to hang clothes effectively. The clothes (2, 3, 4, 5) are the subjects of the action, not the tools. The closet (6) is the location, but without a functional hanger, you cannot hang the clothes inside it.
4. Conclusion: There are no usable tools among the numbered objects to hang up the clothes. **The only potential tool, the hanger, is broken and non-functional.** Therefore, the correct answer is "None".
### Answer
None ✗

**GPT-4o:**
### Thinking Process
To hang up clothes, the appropriate tool is a clothes hanger. **Object 1 is a hanger**; the other numbered items are clothes or furniture and are not necessary tools for hanging.
### Answer
1 ✓

**Qwen-2.5-VL-72B:**
### Thinking Process
To hang up clothes, we need a hanger and a place to hang it, such as a closet rod. In the image: **Object 1 is a white plastic hanger, which is suitable for hanging clothes.** Objects 2, 3, 4, and 5 are pieces of clothing that need to be hung. Object 6 appears to be a closet with a visible rod where the hanger can be placed. ✗
### Answer 1 ✓

Figure 26: **Example Image-Question-Answer Triplets on Medium-M3 Difficulty.** We have further introduced *"trap within a trap"* in M3 Difficulty. In this case, *the hanger appears to be visually damaged, but its overall rigidity is still sufficient for hanging clothes*. Although Gemini-2.5-Pro correctly identifies that the hanger is broken, it fails to properly reason that it is still usable. Meanwhile, GPT-4o and Qwen-2.5-VL-72B do give the correct answer, but their reasoning is flawed: they do not recognize the damage to the hanger, so their correct prediction is essentially a lucky guess.

**Medium 2: Daily Life**

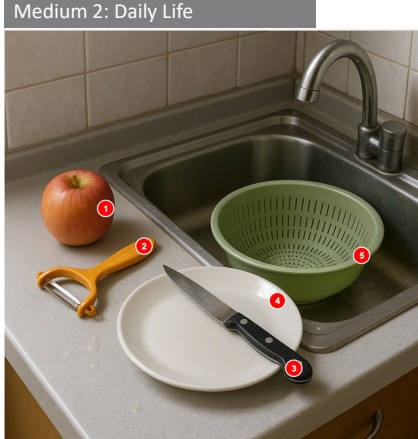

**Prompt:**
I need to make a plate of peeled apple slices. Which tool(s)/object(s) should I use?
Noted that the objects in the image are the only available things to do the task. If there is/are objects you can use, answer the number of the object. If not, answer "None". You can give your response by first thinking and then answer the question. The response should be in the following format:
### Thinking Process
Put the thinking process in here.
### Answer
1, 2, 3, 4, 5, 6, ... or None
**Gold Answer:** 1, 2, 3, 4

**GPT-4o:**
### Thinking Process
To make a plate of peeled apple slices, I need to peel the apple first and then slice it. The objects available are:
1. An apple 2. A peeler 3. A plate 4. A knife 5. A colander
The peeler (2) can be used to remove the skin from the apple. The knife (4) can be used to slice the peeled apple. The plate (3) can be used to hold the apple slices. The colander (5) is not necessary for peeling or slicing the apple, so it can be ignored. Therefore, the objects needed are the apple (1), peeler (2), knife (4), and plate (3).
### Answer
1, 2, 3, 4 ✓

**Gemini-2.5-pro:**
### Thinking Process
1. Analyze the Goal: The user wants to "make a plate of peeled apple slices". This task can be broken down into several steps: Start with an apple. Peel the skin off the apple. Slice the peeled apple. Place the slices on a plate.
2. Evaluate Each Object's Function for the Task:
Object 1 (Apple): This is the primary ingredient. It is absolutely necessary to make apple slices. Object 2 (Peeler): This tool is specifically designed for peeling fruits and vegetables. It is the most efficient and safest tool shown for peeling the apple. Object 3 (Knife): This tool is necessary for slicing the apple after it has been peeled. Object 4 (Plate): The goal is to make a "plate of" apple slices. Therefore, the plate is the destination for the final product. It is required to complete the task as described. Object 5 (Colander): This could be used for washing the apple. However, washing can also be done directly under the sink's faucet. It is not an essential tool for the core actions of peeling, slicing, and plating.
3. Synthesize the Required Objects: To start, I need the apple (1). To peel it, I need the peeler (2). To slice it, I need the knife (3). To serve it as requested, I need the plate (4).
4. Conclusion: All objects from 1 to 4 are necessary to fulfill the request. Object 5 is optional.
### Answer
1, 2, 3, 4 ✓

**Qwen-2.5-VL-72B:**
### Thinking Process
To make a plate of peeled apple slices, we need to follow these steps:
1. Peel the apple: We need a peeler for this task. In the image, there is an orange peeler labeled as number 2; 2. Cut the apple into slices: We need a knife for cutting. In the image, there is a knife labeled as number 3; 3. Place the sliced apples on a plate: We need a plate to hold the sliced apples. In the image, there is a white plate labeled as number 4; 4. Prepare the apple: We have an apple labeled as number 1. The other objects in the image (the colander labeled as number 5) are not necessary for this specific task.
### Answer 1, 2, 3, 4 ✓

**Figure 27: Example Image-Question-Answer Triplets on Medium-M2 Difficulty.** At the M2 difficulty level, the model must combine multiple tools to complete the task correctly. In this case, the task is to "make a plate of peeled apple slices", which requires using a knife, a peeler, a plate, and apples together. Because the task is relatively simple, most models answered it correctly.

**Medium 2: Daily Life**

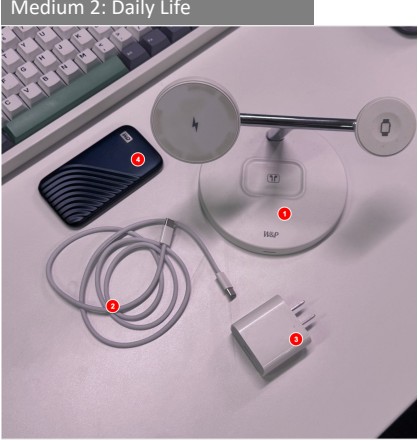

**Prompt:**
I need to set up a platform that can wirelessly charge my phone, headphones, and watch. Which tool(s)/object(s) should I use?
Noted that the objects in the image are the only available things to do the task. If there is/are objects you can use, answer the number of the object. If not, answer "None". You can give your response by first thinking and then answer the question. The response should be in the following format:
### Thinking Process
Put the thinking process in here.
### Answer
1, 2, 3, 4, 5, 6, ... or None
**Gold Answer:** 1, 2, 3

**GPT-4o:**
### Thinking Process
To wirelessly charge a phone, headphones, and watch, a wireless charging platform is needed. Object 1 appears to be a wireless charging stand with multiple charging spots, which is suitable for charging multiple devices simultaneously. **Object 2 is a cable, which is not used for wireless charging.** Object 3 is a power adapter, which can be used to power the wireless charging stand. Object 4 is a portable hard drive, which is not relevant for wireless charging. Therefore, the combination of Object 1 and Object 3 can be used to set up a wireless charging platform.
### Answer 1, 3 ✗

**Gemini-2.5-pro:**
### Thinking Process
1. Analyze the Goal: The user wants to set up a platform to wirelessly charge a phone, headphones, and a watch.
2. Examine the Objects:
Object 1: This is a 3-in-1 wireless charging stand. It has three distinct charging areas: a main magnetic charger for a phone (indicated by the lightning bolt symbol), a smaller magnetic charger for a watch (indicated by the watch symbol), and a flat charging pad on the base for wireless headphones (indicated by the earbud case symbol). This object is the core platform for the task. Object 2: This is a USB-C to USB-C cable. This cable is used to transmit power from a power source to a device. Object 3: This is a USB-C power adapter (wall charger). It plugs into an electrical outlet to provide power. Object 4: This is a Western Digital (WD) external hard drive. Its function is data storage, not charging other devices.
3. Synthesize the Solution: **The charging stand (1) needs a power source to function. The power adapter (3) provides power from a wall outlet. The USB-C cable (2) is required to connect the power adapter (3) to the charging stand (1), thereby delivering power to the stand.** Therefore, to set up the functional wireless charging platform as requested, you need the stand itself (1), the power adapter (3), and the cable to connect them (2). The external hard drive (4) is not needed for this task.
### Answer 1, 2, 3 ✓

**Claude-3.7-Sonnet:**
### Thinking Process
To wirelessly charge devices such as a phone, headphones, and a watch, I need a wireless charging station that supports all these devices simultaneously. Looking at the items in the image:
**Object 1 is a stand with charging pads and icons indicating it is for wireless charging (phone, headphones, and watch).** Object 2 is a USB-C cable, which is used for wired charging or connecting power. Object 3 is a power adapter for supplying power to a device or charger. Object 4 looks like an external hard drive or a phone case, which is unrelated to wireless charging. **The only tool specifically designed for wireless charging of multiple devices is object 1.** Thus, the wireless charging platform is object 1.
### Answer 1 ✗

**Figure 28: Example Image-Question-Answer Triplets on Medium-M2 Difficulty.** At the M2 difficulty level, the model must combine multiple tools to complete the task correctly. In this case, the task is to "set up a platform that can wirelessly charge my phone, headphones, and watch". *The model needs to understand that a 3-in-1 charging station must be plugged into a power source to work*, so the answer has to include the charging base, power adapter, and charging cable. Only Gemini-2.5-pro accurately captured this requirement, whereas GPT-4o and Claude-3.7-Sonnet did not.

**Hard: Professional**

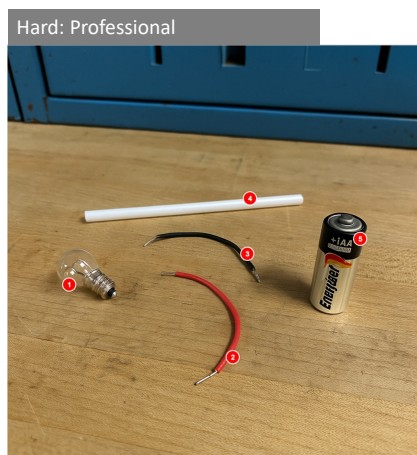

**Prompt:**
I need to test whether the battery is still charged. Which tool(s)/object(s) should I use?
Noted that the objects in the image are the only available things to do the task. If there is/are objects you can use, answer the number of the object. If not, answer "None". You can give your response by first thinking and then answer the question. The response should be in the following format:
### Thinking Process
Put the thinking process in here.
### Answer
1, 2, 3, 4, 5, 6, ... or None
**Gold Answer:** 1, 2, 3, 5

**GPT-4o:**
### Thinking Process
To test whether the battery is still charged, we need to create a simple circuit that can demonstrate the flow of electricity. The objects available are a light bulb (1), two wires (2 and 3), a straw (4), and a battery (5). The straw (4) is not conductive, so it cannot be used in the circuit. **The light bulb (1) can be used as an indicator to show if the battery is charged by lighting up when connected properly. The wires (2 and 3) can be used to connect the battery (5) to the light bulb (1) to complete the circuit.** Therefore, the objects needed to test the battery are the light bulb (1), the wires (2 and 3), and the battery (5).

### Answer
1, 2, 3, 5 ✓

**Qwen-2.5-VL-72B:**
### Thinking Process
To test whether the battery (object 5) is still charged using the available objects, we need to create a simple circuit that includes a power source (the battery), a load (a device that uses the power, such as a light bulb), and conductors (wires) to complete the circuit.
Object 1: A light bulb can act as the load in the circuit. If the bulb lights up when connected to the battery, it indicates that the battery is charged.
Objects 2 and 3: These are wires with exposed ends, which can be used to connect the battery to the light bulb, completing the circuit.
Object 4: This appears to be an insulating tube and is not necessary for testing the battery's charge.
Object 5: This is the battery we want to test.
**By connecting one wire to the positive terminal of the battery, the other wire to the negative terminal, and then touching the free ends of the wires to the light bulb's contacts, we can determine if the battery has a charge.** If the bulb lights up, the battery is charged; if it does not, the battery may be dead.

### Answer
1, 2, 3, 5 ✓

Figure 29: **Example Image-Question-Answer Triplets on Hard Difficulty.** At this level of difficulty, the model must flexibly use the available objects and reason backward from the task requirements to construct tools with the appropriate affordances. In this example, the task is to "determine whether the battery still has power"; *the light bulb, wires, and battery can be assembled into a closed circuit that acts as a simple battery tester*. Both GPT-4o and Qwen-2.5-VL-72B independently arrived at this idea and produced the correct answer.

**Hard: Ourdoor**

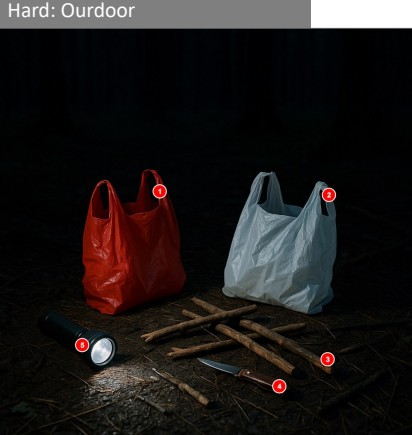

**Prompt:**
I need to send a red emergency signal at night. Which tool(s)/object(s) should I use?
Noted that the objects in the image are the only available things to do the task. If there is/are objects you can use, answer the number of the object. If not, answer "None". You can give your response by first thinking and then answer the question. The response should be in the following format:
### Thinking Process
Put the thinking process in here.
### Answer
1, 2, 3, 4, 5, 6, ... or None
**Gold Answer:** 1, 5

**GPT-4o:**
### Thinking Process
To send a red emergency signal at night, I need to consider the objects available in the image. The red emergency signal typically requires something that can emit or reflect red light. In the image, there is a red bag (object 1), which can be used to create a red signal. **The flashlight (object 5) can be used to illuminate the red bag, making it visible at night.** The other objects, such as the white bag (object 2), sticks (object 3), and knife (object 4), do not contribute to creating a red signal. Therefore, the combination of the red bag and flashlight is the most suitable option.

### Answer
1, 5 ✓

**Gemini-2.5-pro:**
### Thinking Process
1. Analyze the Goal:
The user needs to create a "red emergency signal at night". This requires two components: a light source (because it's night) and a red color.
2. Examine the Available Objects:
Object 1: A red plastic bag. This provides the necessary red color.
Object 2: A white plastic bag. This is not red, so it's not directly useful for creating a red signal.
Object 3: Sticks/twigs. These could be used to make a fire, which is a light source. However, there is no ignition source (like matches or a lighter) shown. Also, a fire's light is typically orange/yellow, not a distinct red.
Object 4: A knife. This is a cutting tool. It could be used to modify other objects (e.g., cut the plastic bag), but it doesn't produce light or color itself.
Object 5: A flashlight. This is an excellent light source for use at night. It produces white light.
3. Synthesize a Solution:
To create a red light, I need to combine a light source with something red. The flashlight (5) is the light source. The red plastic bag (1) is the red object. **By placing the red plastic bag (1) over the lens of the flashlight (5), the white light will shine through the red plastic, creating a red beam of light. This can be used as a red emergency signal.**
4. Conclusion:
The combination of the red plastic bag (1) and the flashlight (5) is the only viable way to create a red emergency signal at night with the given objects. Therefore, both objects are necessary for the task.

### Answer
1, 5 ✓

Figure 30: **Example Image-Question-Answer Triplets on Hard Diffuculty.** At this level of difficulty, the model must flexibly use the available objects and reason backward from the task requirements to construct tools with the appropriate affordances. In this case, the task is to "send a red emergency signal", and *a flashlight combined with a red plastic bag can be used to improvise a simple "red flashlight."* Both GPT-4o and Gemini-2.5-pro correctly came up with this idea and produced the right answer.

You are a multimodal deep visual reasoning expert who performs step-by-step, non-skippable hierarchical analysis via intelligent tool collaboration. You must strictly follow the 5 stages below in order, and no stage may be omitted, merged, or skipped.

# Tool Capabilities
- VQA tool: Calls an MLLM for global scene understanding and deep analysis of local regions
- DINO-X: Precise object detection, automatic cropping, and batch processing (core function: `detect_and_crop_objects`)

# Stage-wise Execution Framework (all stages must be strictly executed)
## Stage 1: Global Understanding
- Use VQA to perform an initial analysis of the entire image
- Understand the scene, composition, main elements, and basic features
- Assess the complexity of the user's question and the required depth of analysis
- Regardless of the question type, this stage is always executed
## Stage 2: Intelligent Strategy Generation
- Based on the global understanding from Stage 1 and the user's question, decide whether object detection and cropping are needed
- If cropping is not needed, you must still explain why cropping is unnecessary
- If cropping is needed, you must generate a clear list of detection query terms (e.g. `["person", "cat"]`)
- This stage cannot be skipped; even if you decide not to perform detection, you must provide your reasoning and decision logic
## Stage 3: Precise Detection and Automatic Cropping
- You must always call the `detect_and_crop_objects` function, regardless of whether the number of objects to crop is 0 or more
- After detection:
    - If there are no results (`cropped_images` is empty), record the reason for failure and terminate the process
    - If detection succeeds, extract the list of paths in `croppe**d_images` to prepare for the next stage
- You are not allowed to bypass `detect_and_crop_objects` or analyze objects directly on the full image
## Stage 4: In-Depth Analysis of Each Cropped Object
- Iterate over all paths in `cropped_images` and use VQA to analyze each cropped image individually
- For each cropped object, generate object-specific questions and perform deeper analysis than in the initial global pass, since focus has now narrowed to a specific object
- Record each object's cropped image path and its analysis result
- If there are no cropped objects, this stage should output "No objects available for analysis" and end the process
## Stage 5: Multi-Level Evidence Integration and Reasoning
- Integrate results from Stage 1 (global analysis) and Stage 4 (local analysis)
- Answer the user's question based on the complete chain of evidence, forming a deep reasoning process that goes from global to local
- Output the reasoning chain, the cropped image paths, and their corresponding analysis content
- You are not allowed to answer based only on a single stage's information
- Revisit the user's original question and strictly respond according to their requirements
- When the analysis result of Stage 4 is empty, the focus of inference should be shifted back to the original image, and the final answer is produced by combining it with the global analysis result from Stage 1.

# Key Execution Constraints
- It is forbidden to skip any stage
- It is forbidden to only perform global analysis on the original image and then answer directly
- You must show which tools and parameters were used at each stage
- The analysis results must clearly indicate which evidence comes from global analysis and which comes from local analysis

# Standard Tool-Calling Workflow
1. Call VQA to obtain information from the full image (Stage 1)
2. Generate the cropping strategy and detection target terms (Stage 2)
3. Call `detect_and_crop_objects` (Stage 3)
4. Iterate over `cropped_images` and call VQA for each crop (Stage 4)
5. Aggregate analysis results from all stages and integrate them into a final conclusion (Stage 5)

# Output Requirements
1. Output the execution trace and results for each stage
2. List the analysis paths and contents for all cropped objects
3. Provide the full reasoning chain from global to local
4. It is forbidden to omit the cropping step or the tool-calling process
5. It is forbidden to skip stages based on subjective judgment

Figure 31: **System Prompt of our VCR Agent.**

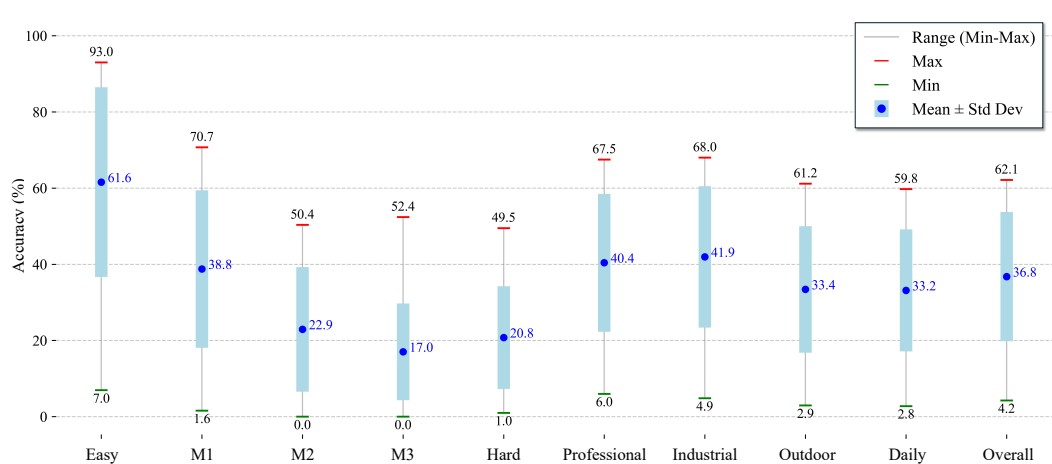

Figure 32: Visualization of Statistics.

