# OpenReview forum: "PhysToolBench: Benchmarking Physical Tool Understanding for MLLMs"
_ICLR.cc/2026/Conference — Submitted to ICLR 2026_

### Official Review · Reviewer_nNjS · 2025-11-01

**Soundness:** 3
**Presentation:** 2
**Contribution:** 2
**Rating:** 4
**Confidence:** 4

**Summary:**

This paper introduces *PhysToolBench*, a visual question answering (VQA) benchmark designed to assess multimodal LLMs’ understanding of physical tools across realistic, resource-constrained scenes. Each example presents a task description and a 1024×1024 image with numerically labeled objects; models must select the applicable tools or answer None. The benchmark spans four domains (Daily Life, Industrial, Outdoor, Professional) and three difficulty tiers: Easy (tool recognition), Medium with subtracks named M1 (attribute understanding), M2 (tool combination), M3 (availability/viability), and Hard (creative tool creation/substitution). Model evaluation shows a wide gap to human performance and highlights especially poor robustness on availability and long-tail categories. The paper further proposes a vision-centric reasoning (VCR) agent that decomposes reasoning via object detection and localized analysis to improve M3 scores.

**Strengths:**

1. The paper is well written and easy to follow.
2. The benchmark shifts tool use evaluation from generic affordance Q&A toward task-conditioned tool selection under availability constraints, which better mirrors embodied settings.
3. A broad model suite (32 MLLMs) across proprietary, open-source, embodied-specific, and VLA-backbones enables nuanced comparisons.

**Weaknesses:**

1. Around 90% of images are generated (GPT-4o-image) rather than captured in real homes/worksites. While curated, this may under-represent clutter, occlusion, wear, and domain shift typical of robotics. It would help to quantify the 10% real photos’ distribution and to report performance gaps between synthetic vs. real subsets. Since text prompts sometimes hinge on brand/model specifics, generated scenes risk subtle stylistic cues that models could exploit. A leakage/stylization analysis would further strengthen the claims.
2. For M2 Tool Combination, success may depend on understanding hidden state not visually disambiguated. To clarify on how such requirements are visually conveyed and how answer correctness is adjudicated would help.
3. The proposed VCR pipeline is only evaluated on M3 with two backbones (GPT-4o/5). It’s promising, but without broader coverage (M1/M2/Hard), latency/runtime overheads, or robustness checks (e.g., detector errors), it’s hard to gauge its generality or practicality in agents.
4. The approach relies on an object detector (DINOX) and crops. But the performance may hinge on detector recall for small/occluded tools. The paper should further quantify detector metrics and report end-to-end sensitivity to detection failure.

**Questions:**

1. Could you report per-subset results (synthetic vs. photographed), to help readers better assess the benchmark's real-world transfer?
2. How do you treat cases with multiple acceptable substitutes or partially adequate tools?
3. Beyond evaluation, have you explored training with VCR-style intermediate supervision (crop-level rationales, attribute labels, or damage present tags)? A small tool-centric visual-reasoning dataset might turn your agent into a method contribution, not only a benchmark.
4. Typo: The LaTeX quotes are incorrect—use the backtick ` for the left quote, not the curly apostrophe '.

---

> ### Author Response · Authors · 2025-11-20
> **Response to Reviewer nNjS (1/3)**
>
> > **W.1:** Around 90% of images are generated (GPT-4o-image) rather than captured in real homes/worksites. While curated, this may under-represent clutter, occlusion, wear, and domain shift typical of robotics. It would help to quantify the 10% real photos’ distribution and to report performance gaps between synthetic vs. real subsets.
> >
> > Since text prompts sometimes hinge on brand/model specifics, generated scenes risk subtle stylistic cues that models could exploit. A leakage/stylization analysis would further strengthen the claims.
> >
>
> **A.1:** First, as described in Section B.2 of the supplementary material, both MLLM-based and human evaluations indicate that the images are highly realistic. Besides, we want to clarify that our benchmark is mainly designed to test MLLMs' understanding of physical tools, and we believe that this is relatively independent and orthogonal from evaluating the models in some challenging real-world scenarios including clutter, occlusion, and wear.
>
> In addition, thanks for the reviewer's insightful suggestion, we provide per-subset results here in **A.5**.
> As shown in there, the success rates on real-world scenarios and synthetic scenarios are quite close, which provides preliminary evidence that the gap between the two is relatively small.
>
> Regarding potential leakage, we would like to clarify our dataset construction pipeline in more detail: 1. Human experts come up with a **task**(e.g. cut the potato) and corresponding **scene description**(e.g. On the kitchen counter, there is a knife, a potato, ...). 2. **Only the scene description** is provided to GPT-4o for generation.
> In other words, during the image generation process, GPT-4o is simply rendering the scene description in a largely mechanical way, without access to or awareness of the specific task the image is intended to support.
> Whether MLLMs can correctly identify the appropriate tool based on a complex task description entirely depends on its reasoning ability and its “understanding” of these tools.
>
> > **W.2:** For M2 Tool Combination, success may depend on understanding hidden state not visually disambiguated. To clarify on how such requirements are visually conveyed and how answer correctness is adjudicated would help.
> >
>
> **A.2:** We have already provided critical information in the images, and the task descriptions are made as detailed as possible to avoid ambiguity. And the high human-test accuracy further demonstrates that this benchmark does not suffer from significant ambiguity (otherwise, the human score will also be low) . We additionally provide several representative cases in Fig.27 and Fig.28 in the updated version of the paper for reference.

---

> ### Author Response · Authors · 2025-11-20
> **Response to Reviewer nNjS (2/3)**
>
> > **W.3:** The proposed VCR pipeline is only evaluated on M3 with two backbones (GPT-4o/5). It’s promising, but without broader coverage (M1/M2/Hard), latency/runtime overheads, or robustness checks (e.g., detector errors), it’s hard to gauge its generality or practicality in agents.
> >
>
> **A.3:** We would first like to emphasize that our current method is intended only as a **baseline** to demonstrate the importance of reasoning, especially *visual* reasoning, for the correct use of physical tools. In this sense, the baseline is mainly designed as a proof of concept and a starting point to encourage future work, rather than a fully optimized solution. However, we are grateful for the reviewers’ rigorous assessment, and we therefore provide further information on the VCR method below. (They are also included in the updated PDF)
>
> **Full Results of VCR.** Following the reviewer’s suggestion, we have added the corresponding results. Specifically, we now report the full performance of our method with VCR across all difficulty levels. These experiments are conducted on top of our strongest backbone, GPT-5, and we systematically evaluate the effect of adding VCR.
>
> | **Method** | **Easy** | **Medium-M1** | **Medium-M2** | **Medium-M3** | **Hard** | **Total** |
> | --- | --- | --- | --- | --- | --- | --- |
> | GPT-5 | 90.16 | 63.83 | 50.35 | 36.75 | 46.04 | 62.51 |
> | +VCR | 89.52 | 66.49 | 51.06 | 54.82 | 47.52 | 65.81 |
>
> We observe that accuracy improves further at most difficulty levels when VCR is incorporated. The improvement is most pronounced at the M3 level, where visual reasoning is most critical. For the other difficulty levels, accuracy increases slightly, whereas at the easiest level we see a small performance drop. We believe this is because the easiest level relies less on complex reasoning, so introducing additional intermediate steps can sometimes accumulate errors and thus slightly degrade the final decision.
>
> **Latency/runtime profile.** Since both the MLLM backbone in our agent (GPT-5) and the object detection model (DINO-X) are closed-source and only accessible via APIs, the runtime is heavily influenced by network conditions and the current load on the model servers. Compared with the original approach (which requires only 1 $\times$ GPT), our method inevitably introduces multiple API calls (1 $\times$ DINO-X, $(n+2)$ $\times$ GPT), where $n$ is the number of objects that require deeper analysis. But these $n$ API calls can be parallelized, so the overall inference time is only about 3-4 times that of the normal mode.
>
> > **W.4:** The approach relies on an object detector (DINOX) and crops. But the performance may hinge on detector recall for small/occluded tools. The paper should further quantify detector metrics and report end-to-end sensitivity to detection failure.
> >
>
> **A.4:** First, we would like to clarify that our dataset does not contain any ground-truth detection results.
>
> Second, regarding sensitivity to detection failures, we have explicitly taken this into account in the agent’s system prompt(provided in Fig.31 in the updated PDF). The analysis performed after the Detect+Crop step is treated only as auxiliary information; the final reasoning is always carried out with reference to the original image.
>
> Moreover, we would like to reiterate that this baseline is mainly intended to show that more focused visual reasoning indeed improves tool understanding, whereas the primary focus of our work is the benchmark itself.

---

> ### Author Response · Authors · 2025-11-20
> **Response to Reviewer nNjS (3/3)**
>
> > **Q.1:** Could you report per-subset results (synthetic vs. photographed), to help readers better assess the benchmark's real-world transfer?
>
> **A.5:** This is a very insightful suggestion, and we now provide the corresponding results.
>
> As shown in the table below, the success rates on real-world scenarios and synthetic scenarios are quite close, which provides preliminary evidence that the gap between the two is relatively small.
> | **Model** | **Synthetic** | **Real** | **Total** |
> | --- | --- | --- | --- |
> | GPT-4o | 62.07% | 54.29% | 61.26% |
> | Gemini-2.5-pro | 54.80% | 51.43% | 54.45% |
> | Claude-3.7-Sonnet | 51.05% | 46.67% | 50.59% |
> | GLM-4.5-V-108B | 55.13% | 55.24% | 55.14% |
> | Qwen-2.5-VL-72B | 49.28% | 51.43% | 49.51% |
>
>
> > **Q.2:** How do you treat cases with multiple acceptable substitutes or partially adequate tools?
>
> **A.6:** All of our questions have undergone multiple rounds of expert review to ensure that the annotated answer is indeed the best choice. For partially adequate tools, the correct answer should be **None**, because we explicitly inform the model that the tools shown in the image are the only tools available. Given the condition that the objects in the image are the only available tools, the answer should be **None** if the tool is partially adequate. For example, in Fig.23, although the remote control seems to have the buttons to turn on the TV, it is actually not adequate to complete the task because it lacks batteries. Therefore, the correct answer should be **None**.
>
>
> > **Q.3:** Beyond evaluation, have you explored training with VCR-style intermediate supervision (crop-level rationales, attribute labels, or damage present tags)? A small tool-centric visual-reasoning dataset might turn your agent into a method contribution, not only a benchmark.
>
> **A.7:** This is a very insightful suggestion, and we agree that it points to a promising direction for future work. However, the main focus of this paper is the benchmark itself. Our PhysToolBench is designed to *quantitatively* evaluate multimodal large models’ understanding of physical tools—a problem that, to the best of our knowledge, has not been systematically studied before. We hope that our work will help move this area forward.
>
> We believe the suggested approaches could serve as strong solution candidates for our benchmark. However, our goal for PhysToolBench is to establish a guiding benchmark—a reference point that can inspire and steer future research on how MLLMs and agents understand and use physical tools.
>
> > **Q.4:** Typo: The LaTeX quotes are incorrect—use the backtick ` for the left quote, not the curly apostrophe '.
>
> **A.8:** We thank the reviewer for pointing this out. We have thoroughly revised the manuscript and uploaded an updated version.

---

> ### Author Response · Authors · 2025-11-28
> **Looking forward to further discussion**
>
> Dear Reviewer nNjS,
>
> We would like to express our sincere thanks for the time and effort you have devoted to the reviewing process. To facilitate a clearer understanding of our rebuttal and revisions, we summarize your main concerns and our corresponding responses as follows:
>
> - **On real-world transfer (`weakness1 & question1`).** The reviewer pointed out that 90% of the images in our benchmark are generated. In the rebuttal, we compared results across different subsets (real-world & generated) and found that the performance differences are minor, suggesting that our benchmark does not suffer from a significant domain gap.
> - **Possible misunderstanding on ambiguity in benchmark design (`weakness2 & question2`).** The reviewer raised concerns about potential ambiguity in our benchmark. We make clarification in A.2 and A.6 of the rebuttal. Our benchmark underwent multiple rounds of internal review to minimize ambiguity as much as possible. Moreover, in the design, we explicitly state that the tools shown in the images are the only available objects. We also include illustrative cases in Fig. 27 and Fig. 28 of the revised manuscript to further clarify this point.
> - **On additional details and results for the proposed VCR (`weakness3 & weakness4 & question3`).** We provide additional results for GPT-5 + VCR as well as more implementation details in A.3 and A.4 of the rebuttal, and in Section D of the supplementary material of the revised manuscript. We appreciate the reviewer’s careful suggestions. We would also like to emphasize that the main contribution of our work is introducing the **first benchmark for physical tool understanding**; the VCR component is primarily intended to support one of our key findings—namely, that visual-level reasoning plays a crucial role.
>
> For other issues not mentioned here and more details, please kindly refer to our detailed rebuttal response.
>
> **We sincerely hope this addresses your concerns, and we very much look forward to any further discussion.**
>
> Warm regards,
>
> Authors

---

### Official Review · Reviewer_RBRq · 2025-11-01

**Soundness:** 3
**Presentation:** 3
**Contribution:** 2
**Rating:** 4
**Confidence:** 4

**Summary:**

This paper introduces PhysToolBench, the benchmark designed to evaluate how well Multimodal Large Language Models (MLLMs) comprehend physical tools. Structured as a Visual Question Answering (VQA) dataset with over 1,000 image-text pairs , its primary contribution is a three-level evaluation framework assessing Tool Recognition (basic function) , Tool Understanding (attributes, combination, and availability) , and Tool Creation (flexible usage). By testing 32 MLLMs, the authors demonstrate a significant deficiency in current models , finding that even the best score no higher than 63% (compared to human performance over 90%) and revealing critical weaknesses, especially a failure to recognize when tools are damaged or non-functional. The paper also provides an in-depth analysis of these failings and proposes a "vision-centric reasoning" framework as a preliminary solution.

**Strengths:**

- Definition of a new problem: While many benchmarks test LLMs on digital tools (like API calls) or MLLMs on simple affordance recognition (i.e., "what is this for?") , this paper is the first to create a dedicated benchmark to evaluate the practical, deep understanding of physical tools.
- The experimental quality is a major strength. The authors conducted a massive evaluation of 32 MLLMs.
- The paper is well written and easy to follow.

**Weaknesses:**

- The benchmark's design, while original, has limitations in what it truly evaluates. Its core weakness is the potential confusion between deep physical reasoning and visual perception combined with "common sense" world knowledge.
  - This is most evident in the "Availability Understanding" (M.3) sub-task. The paper highlights models' failures on this task as a critical finding. However, this task primarily tests the model's ability to perceive visual damage (e.g., "cracked," "damaged," "empty") and associate it with a learned fact ("cracked things are unusable"). This does not necessarily validate that the model understands the underlying physical principles of why the tool fails (e.g., why a plunger requires an intact seal to create a pressure differential). The benchmark's scope is thus limited in its ability to truly test the comprehension of physics.
  - The "Tool Creation" (Lv.3) level, while innovative, is constrained by the VQA format, which implies a single correct answer. In reality, "tool creation" (or more accurately, tool substitution) is an open-ended problem. For a task like "tighten a screw," a coin might be the ground truth, but a nearby metal ruler, a butter knife, or a different flat-edged object could also be valid solutions. The paper does not address how the benchmark handles this evaluation ambiguity or credits multiple correct, creative solutions.
  - The provided images, while high-quality, appear to be heavily staged and simplified. The paper notes that items are numerically labeled and are the "only available things". This is a significant simplification of a true "open-world" scenario. A real environment (e.g., a kitchen, a workshop) is cluttered, and the correct tool may be partially occluded, surrounded by many similar-looking distractors, or require searching (e.g., in a drawer).
- The paper's contribution, while significant, is centered almost entirely on the introduction of a new dataset and the resulting analysis, with less emphasis on a novel methodological advancement. The paper's scope and structure align more with the goals of a benchmark track than the main conferences

**Questions:**

See weaknesses.

---

> ### Author Response · Authors · 2025-11-20
> **Response to Reviewer RBRq (1/3)**
>
> > **Q.1:** The benchmark's design, while original, has limitations in what it truly evaluates. Its core weakness is the potential confusion between deep physical reasoning and visual perception combined with "common sense" world knowledge.
>
> **A.1:** There might be some misunderstanding related to our benchmark due to the limited scope of case we can show in our paper. For each of the misunderstandings, we have provided detailed clarification below.
> ***We kindly invite the reviewer to examine the response below and the additional cases in the updated PDF, which we hope will clarify the design of our benchmark and address the misunderstandings.***
>
> > **Q.1.1:** This is most evident in the "Availability Understanding" (M.3) sub-task. The paper highlights models' failures on this task as a critical finding. However, this task primarily tests the model's ability to perceive visual damage (e.g., "cracked," "damaged," "empty") and associate it with a learned fact ("cracked things are unusable").
>
> **A.1.1:** We respectfully disagree with this statement of "this task primarily tests the model's ability to perceive visual damage". In fact, in the M3-level *Availability Understanding* task, **we are not simply applying “visual damage” to the objects. Instead, through careful design, we make the core functional components of a tool unavailable or unsuitable for the task**, so that the tool seems to be the right one but in reality cannot accomplish the task.
>
> For example, in the TV remote shown in Fig.6.d (full question–answer pair for this example in the updated PDF in Fig.23), the only abnormal situation is that battery cover is open and shows the lack of batteries. **It is therefore inaccurate to say that the remote has “visual damage”**. A model cannot simply rely on a shallow heuristic like “cracked things are unusable” to answer this question. Rather, it must engage in deeper reasoning and understand that a remote control requires batteries to function. **We also include additional cases in Fig.24.** In one of them, the task is to measure an angle, but the scale on the protractor is missing. From the outside, the protractor itself looks completely intact; however, a model must understand that the scale is essential for measuring angles in order to answer the question correctly. Through such carefully designed cases, we can more directly test a model’s genuine understanding of why a tool is or is not usable.
>
> In addition, our M3 difficulty level includes a diverse range of cases. For example, **we may provide a tool that is functionally similar to the required one but actually unsuitable for the task**. As shown in Fig.25, the task is to precisely measure 5 grams of flour, but the only tool given is a bathroom scale. The model must realize that the scale’s precision is insufficient and therefore answer “None.” Cases like this cannot be solved simply by applying a trivial heuristic such as “broken tools cannot be used.”
>
> Moreover, we actually further introduce **a deeper layer of traps** (a “trap within a trap”): items that appear to have visual damage, but whose core functionality is actually intact, and thus remain usable. These cases test whether a model truly understands *why* a tool still works despite superficial damage. Somewhat ironically, most current models generally fail even at proactively analyzing whether a tool is damaged, let alone determining that “despite the damage, it is still usable,” and therefore only answer such questions correctly by chance. Nevertheless, as a benchmark, we deliberately retain these cases so that, as more capable models and more powerful reasoning paradigms emerge, our dataset will continue to serve as a meaningful and challenging evaluation.
>
> In summary, the core principle behind the design of our M3 difficulty level is that we make the core functional components of a tool unavailable or unsuitable for the task, and this can take many forms, such as physical damage to the tool, a mismatch between the tool’s functionality and the task, or the absence of key components. In addition, we introduce “trap within a trap” constructions. The low scores of current large models on the M3 level also indicate that their understanding of tools is relatively shallow: they tend to rely on memorized correspondences between simple tools and tasks rather than genuine understanding.

---

> ### Author Response · Authors · 2025-11-20
> **Response to Reviewer RBRq (2/3)**
>
> > **Q.1.2:** The "Tool Creation" (Lv.3) level, while innovative, is constrained by the VQA format, which implies a single correct answer. In reality, "tool creation" (or more accurately, tool substitution) is an open-ended problem. For a task like "tighten a screw," a coin might be the ground truth, but a nearby metal ruler, a butter knife, or a different flat-edged object could also be valid solutions. The paper does not address how the benchmark handles this evaluation ambiguity or credits multiple correct, creative solutions.
>
> **A.1.2:** First, regarding the phrase “or more accurately, tool substitution,” **we respectfully disagree with this characterization**. In our Hard setting, there are many cases where the MLLM must use the tools at hand to *create* new tools. For example, in several newly added cases such as Fig. 30, the MLLM must combine a flashlight with a red plastic bag to create a “red flashlight” that can emit red light. In Fig. 29, the model needs to flexibly connect wires and a light bulb to form a simple “voltage tester” in order to check whether the battery still has power. These cases are not merely about substituting one tool for another; instead, the model must reason backward from the task requirements to infer the necessary tool functions, and then use the available objects to realize those functions.
>
> Second, regarding the “open-ended” concern, as stated clearly around Line 188, we explicitly inform the MLLM that the objects in the image are the only usable resources. As shown in Fig.12 to Fig.30, the prompt includes the instruction: “Note that the objects in the image are the only available things to do the task. If there is/are objects you can use, answer the number of the object. If not, answer ‘None’.” This effectively avoids ambiguity by restricting models to selecting from these enumerated objects (or “None”), rather than producing arbitrary free-form responses.
>
> Moreover, all of our ground truth answers have been carefully verified by multiple experts, and the high human-test accuracy further demonstrates that this benchmark does not suffer from significant ambiguity (otherwise, the human score will also be low).

---

> ### Author Response · Authors · 2025-11-20
> **Response to Reviewer RBRq (3/3)**
>
> > **Q.1.3:** The paper notes that items are numerically labeled and are the "only available things". This is a significant simplification of a true "open-world" scenario. A real environment (e.g., a kitchen, a workshop) is cluttered, and the correct tool may be partially occluded, surrounded by many similar-looking distractors, or require searching (e.g., in a drawer).
> >
>
> **A.1.3:** **First, we would like to clarify that our benchmark actually does include scenarios of similar-looking distractors.** For example, as shown in Fig.21 and Fig.22, we introduce distractors that have similar shapes but different materials, specifically to test whether the model truly understands the tools rather than merely relying on superficial appearance.
>
> **Second, regarding occlusion and searching, our benchmark does not cover these cases for the following reasons:**
>
> 1. To the best of our knowledge, no prior work has systematically analyzed MLLMs’ capabilities in *physical tool understanding*. **Our aim is to provide a dedicated, comprehensive benchmark and in-depth analysis focused specifically on this ability**, in order to contribute to the field and help drive further progress.
> 2. **Incorporating too many real-world factors would make it difficult to pinpoint the true sources of model errors.** For instance, we would not be able to tell whether a wrong answer is caused by occlusion or by insufficient recognition and understanding of the physical tools themselves. We therefore deliberately decouple these challenges so that we can more accurately diagnose the models’ fundamental weaknesses.
> 3. **We believe that evaluating understanding of physical tools is relatively independent and orthogonal from evaluating the model in occlusion scenarios and on long-horizon task planning that requires searching.** Meanwhile, there already exist specialized benchmarks regarding occlusion and long-horizon task planning. For example, there are benchmarks for evaluating model performance under occlusion [1], and for long-horizon task planning [2]. Different benchmarks naturally emphasize different aspects. **Our benchmark is designed to focus on MLLMs’ understanding of physical tools and is complementary to those existing benchmarks.** In practice, researchers can combine our benchmark with others according to their specific application needs.
> 4. **Our benchmark can be viewed as a *minimal test*.** We fully acknowledge that real-world scenarios are more complex—for example, with occlusions or objects outside the field of view. However, if a model cannot succeed on our simplified setting, it is unlikely to handle more complex conditions. In this sense, our benchmark serves as a prerequisite check for whether a model is ready to flexibly use physical tools to solve problems in realistic environments. Indeed, our experimental results show that even the strongest current models perform far from human level (the best score is only 62.15).
>
> Taken together, as the first work in this specific area, our benchmark intentionally decouples these challenges. However, we believe that extending it to include occlusion, searching, and other real-world factors is a very promising direction for future work to better align with practical applications.
>
> [1] Liu, Zhaochen, et al. "Beyond the Visible: Benchmarking Occlusion Perception in Multimodal Large Language Models." *arXiv preprint arXiv:2508.04059* (2025).
>
> [2] Chen, Yi, et al. "Egoplan-bench: Benchmarking multimodal large language models for human-level planning." *arXiv preprint arXiv:2312.06722* (2023).
>
>
> > **Q.2:** The paper's scope and structure align more with the goals of a benchmark track than the main conferences
>
> **A.2:** We respectfully note that, unlike NeurIPS, which has separated benchmark submissions into a dedicated track, the ICLR main conference explicitly accepts benchmark papers. To the best of our knowledge, the ICLR main conference actually accepts a non-negligible number of works about benchmark. In 2025 ICLR, at least **11 out of 213 Oral papers** are benchmarks. Some of them are listed below:
>
> [1] Zhuo, Terry Yue, et al. "BigCodeBench: Benchmarking Code Generation with Diverse Function Calls and Complex Instructions." *The Thirteenth International Conference on Learning Representations*.
>
> [2] Kran, Esben, et al. "DarkBench: Benchmarking Dark Patterns in Large Language Models." *The Thirteenth International Conference on Learning Representations*.
>
> [3] Chow, Wei, et al. "PhysBench: Benchmarking and Enhancing Vision-Language Models for Physical World Understanding." *The Thirteenth International Conference on Learning Representations*.
>
> [4] Zhao, Siyan, et al. "Do LLMs Recognize Your Preferences? Evaluating Personalized Preference Following in LLMs." *The Thirteenth International Conference on Learning Representations*.

---

> ### Author Response · Authors · 2025-11-28
> **Looking forward to further discussion**
>
> Dear Reviewer RBRq,
>
> We would like to express our sincere thanks for the time and effort you have devoted to the reviewing process. To facilitate a clearer understanding of our rebuttal and revisions, we summarize your main concerns and our corresponding responses as follows:
>
> - **Possible misunderstanding regarding benchmark design** **(`weakness 1`)**. Because of the limited number of examples we could include in the paper, we were not able to fully convey the breadth, structure, and systematic nature of our benchmark design. This may have led the reviewer to view our benchmark as simpler than it actually is and to raise several concerns. In the rebuttal (A.1.1–A.1.3), we respond to each of these points in detail, and in the revised manuscript we have added corresponding illustrative figures to better explain the design (Fig. 23–Fig. 30). Overall, PhysToolBench was carefully designed with fine-grained structure to progressively and rigorously assess how well large models understand physical tools.
> - **Suitability for ICLR (`weakness2`)**:  We clarified that the ICLR main conference does in fact accept a substantial number of benchmark-focused papers, and we provided several representative examples to support this point in the rebuttal.
>
> For other issues not mentioned here and more details, please kindly refer to our detailed rebuttal response.
>
> **We sincerely hope this addresses your concerns, and we very much look forward to any further discussion.**
>
> Warm regards,
>
> Authors

---

### Official Review · Reviewer_FTcy · 2025-11-02

**Soundness:** 4
**Presentation:** 4
**Contribution:** 3
**Rating:** 6
**Confidence:** 4

**Summary:**

The paper proposed a benchmark for evaluation MLLMs' performance on understanding tool using tasks. This benchmark is designed to metric the MLLM's ability in three hierarchical difficulty levels. The experimental results indicates that this benchmark reveals several bottleneck of current MLLMs in tool understanding, such as model scale, backbone, vision v.s. text, etc. Then the author proposed a new method to help improving the tool understanding ability for MLLMs by more emphasizing the vision information.

**Strengths:**

1. The paper claimed the first benchmark work for physical tool understanding.
2. The paper is clearly narrated. The tables and figures are clear to read.
3. The experiment is conducted based on a large group of current MLLMs, provides convincing conclusion.

**Weaknesses:**

1. Verbal polish: e.g. task of "open a TV" should be "turn on a TV" in Figure 6. Also, incorrect usage of quotation mark throughout the paper.
2. While the experiments evaluated the performance of MLLMs in different difficulty levels designed by the authors. However, to demonstrate that the metric dimensions are sufficiently unique and efficient, the author should also report the MSE of each metric dimension, so that the reader can understand if this dimension is able to largely distinguish different MLLMs.
3. What would be the result if MLLMs are inferencing visually similar but different tool materials? For example, there are 2 forks on the table, one is made of medal, and another is made of plastic. The task is to pick some heavy food material, so the ground truth could only be the medal fork. I assume sometimes the MLLMs might make incorrect decision.

**Questions:**

See weaknesses

---

> ### Author Response · Authors · 2025-11-20
> **Response to Reviewer FTcy**
>
> > **Q.1:** Verbal polish
>
> **A.1:** We thank the reviewer for pointing this out. We have thoroughly revised the manuscript and uploaded an updated version.
>
> > **Q.2:** While the experiments evaluated the performance of MLLMs in different difficulty levels designed by the authors. However, to demonstrate that the metric dimensions are sufficiently unique and efficient, the author should also report the MSE of each metric dimension, so that the reader can understand if this dimension is able to largely distinguish different MLLMs.
>
> **A.2:** There might be a misunderstanding regarding the evaluation metric. Our evaluation follows a VQA setting: each image contains a numerical label, and the MLLM is required to output the correct label. A prediction is counted as correct if and only if it matches the ground truth (GT); otherwise, it is counted as incorrect. The final metric is overall accuracy.
>
> > **Q.3:** What would be the result if MLLMs are inferencing visually similar but different tool materials? For example, there are 2 forks on the table, one is made of medal, and another is made of plastic. The task is to pick some heavy food material, so the ground truth could only be the medal fork. I assume sometimes the MLLMs might make incorrect decision.
>
> **A.3:** This is an insightful question. **In fact, our benchmark does include scenarios of this type**. Specifically, in the Medium-M1 difficulty level, we introduce a large number of cases that include visually similar distractors, including cases where items look almost identical but are made of different materials. The goal is to test whether a model can reason about the distinct physical properties of different tools and select the most appropriate one based on task requirements.
>
> **We have demonstrated several of these cases in the updated PDF, which are shown in Fig. 21 and Fig. 22.** As illustrated there, some of the strongest commercial models handle these cases with ease, whereas smaller open-source models often fail to correctly process them.

---

> > ### Comment · Reviewer_FTcy · 2025-11-22
> > **Q2**
> >
> > Thanks for the calirfication.
> > For Q2, from my understanding, the authors design a series of VQA tasks, and determine if the model's output matches the GT. But is there a statistic that can shows the degree of distinction in the capabilities of different models? For example, in a "easy" task #1, the best model A achieves 90% accuracy, but the worst model B achieves only 50% accuracy, therefore, we can say task #1 is a good task because it can distinguish different models' ability. However, for a "easy" task #2, the best model A achieves 90% accuracy, but the worst model B achieves 80% accuracy, therefore, we can say task #2 is not a good task because it cannot distinguish different models' ability.
> >
> > Therefore, I think it's good to discuss the variance/standard deviation that measures spread of scores across models:
> > High variance -> better discrimination between models
> > Low variance -> most models perform similarly

---

> > > ### Author Response · Authors · 2025-11-22
> > > **Response to Q2: Updated Additional Statistics and Visualization**
> > >
> > > We fully appreciate the reviewer’s point and agree that it is an insightful suggestion. Accordingly, we have incorporated the relevant data into the revised manuscript. Table 4 now summarizes the benchmark results using four key statistics (maximum, minimum, mean, and standard deviation), while Fig. 32 provides a more intuitive visualization of these values. As shown in the table and the figure, ***across different difficulty levels and scenario categories, the gaps between the maximum and minimum scores across models are large and the standard deviations are substantial, indicating that our benchmark can effectively distinguish models with different capability levels.*** For the reviewer’s convenience, we have reproduced Table 4 below. And we kindly suggest reading it together with Fig. 32, which may help provide a more intuitive understanding of these results.
> > >
> > > | Statistics | Easy | M1 | M2 | M3 | Hard | Professional | Industrial | Outdoor | Daily | Overall |
> > > | --- | --- | --- | --- | --- | --- | --- | --- | --- | --- | --- |
> > > | Max | 93.02 | 70.74 | 50.35 | 52.41 | 49.50 | 67.50 | 68.02 | 61.18 | 59.75 | 62.15 |
> > > | Min | 6.98 | 1.60 | 0.00 | 0.00 | 0.99 | 6.00 | 4.86 | 2.94 | 2.78 | 4.25 |
> > > | Mean | 61.58 | 38.77 | 22.93 | 17.03 | 20.76 | 40.41 | 41.95 | 33.41 | 33.16 | 36.78 |
> > > | Std Dev | 24.89 | 20.67 | 16.34 | 12.71 | 13.48 | 18.11 | 18.56 | 16.59 | 16.01 | 16.98 |

---

> > > > ### Comment · Reviewer_FTcy · 2025-11-23
> > > >
> > > > Thanks, the author's reply addressed all my concerns.

---

### Official Review · Reviewer_6cbv · 2025-11-05

**Soundness:** 3
**Presentation:** 3
**Contribution:** 2
**Rating:** 4
**Confidence:** 2

**Summary:**

The paper proposes a benchmark to evaluate the ability of VLMs and VLAs to understand physical tools.

**Strengths:**

The paper conducts a thorough evaluation across a wide range of baselines and designs multiple difficulty levels in its benchmark for more detailed analysis.

**Weaknesses:**

The paper is primarily focused on benchmarking, making it more suitable for a benchmark-oriented venue rather than ICLR.

**Questions:**

1. Is GPT achieving the highest score because the images in the benchmark were generated by GPT itself

---

> ### Author Response · Authors · 2025-11-20
> **Response to Reviewer 6cbv**
>
> > **W.1:** The paper is primarily focused on benchmarking, making it more suitable for a benchmark-oriented venue rather than ICLR.
> >
>
> **A.1:** To the best of our knowledge, the ICLR main conference actually accepts a non-negligible number of work about benchmark. In 2025 ICLR, at least **11 out of 213 Oral papers** are benchmarks. Some of them are listed below:
>
> [1] Zhuo, Terry Yue, et al. "BigCodeBench: Benchmarking Code Generation with Diverse Function Calls and Complex Instructions." *The Thirteenth International Conference on Learning Representations*.
>
> [2] Kran, Esben, et al. "DarkBench: Benchmarking Dark Patterns in Large Language Models." *The Thirteenth International Conference on Learning Representations*.
>
> [3] Chow, Wei, et al. "PhysBench: Benchmarking and Enhancing Vision-Language Models for Physical World Understanding." *The Thirteenth International Conference on Learning Representations*.
>
> [4] Zhao, Siyan, et al. "Do LLMs Recognize Your Preferences? Evaluating Personalized Preference Following in LLMs." *The Thirteenth International Conference on Learning Representations*.
>
> We believe that benchmarks are crucial for guiding research in the right direction. Our PhysToolBench is designed to quantitatively measure multimodal large models' understanding of physical tools, **a task that has not been addressed by anyone before**. We hope that our work will contribute to advancing this field and provide some guidance for future research.
>
> > **Q.1:** Is GPT achieving the highest score because the images in the benchmark were generated by GPT itself?
> >
>
> **A.2:** We would like to clarify that during the image generation process, GPT-4o is simply mechanically visualizing the scene description, without any understanding of the specific task it is intended to support. Here is how our process works: 1. Human experts come up with a **task**(e.g., cut the potato) and a corresponding **scene description**(e.g., On the kitchen counter, there is a knife, a potato, ...). 2. **Only the scene description** is provided to GPT-4o for generation.
>
> Whether the model can correctly identify the appropriate tool based on a complex task description entirely depends on its reasoning ability and its “understanding” of these tools.
>
> Although GPT series may be more familiar with the visual appearance of these images, which may make tool “recognition” easier, the model still fails to show a clear advantage over others on the Easy set. For example, GPT-4o’s score on Easy difficulty (86.03%) is even lower than that of the open-source GLM-4.5V model (90.48%). So we believe the high score of GPT series is because of the physical tool understanding and reasoning ability of the models.
>
> For the reviewer’s convenience, we have extracted the relevant data (performance of some top-tier proprietary and open-source models on our benchmark) from Table 1 of the paper and presented it below.
>
> | **Model** | **Easy** | **Medium-M1** | **Medium-M2** | **Medium-M3** | **Hard** | **Total** |
> | --- | --- | --- | --- | --- | --- | --- |
> | GPT-4o | 86.03 | 70.74 | 48.23 | 35.54 | 44.06 | 61.26 |
> | GPT-5 | 90.16 | 63.83 | 50.35 | 36.75 | 46.04 | 62.51 |
> | o3 | 93.02 | 67.02 | 46.81 | 22.89 | 49.50 | 61.56 |
> | GLM-4.5-V-108B | 90.48 | 65.43 | 36.88 | 16.27 | 35.15 | 55.14|
> | Qwen-2.5-VL-72B | 75.56 | 55.85 | 35.46 | 31.93 | 27.23 | 49.51 |

---

> ### Author Response · Authors · 2025-11-28
> **Looking forward to further discussion**
>
> Dear Reviewer 6cbv,
>
> We would like to express our sincere thanks for the time and effort you have devoted to the reviewing process. To facilitate a clearer understanding of our rebuttal and revisions, we summarize your main concerns and our corresponding responses as follows:
>
> - **Suitability for ICLR (`weakness1`)**: We clarified that the ICLR main conference does in fact accept a substantial number of benchmark-focused papers, and we provided several representative examples to support this point in the rebuttal.
> - **On the advantage of GPT-series models on our benchmark (`question1`)**: During the construction of our benchmark, GPT-4o was used solely to convert textual scene descriptions into images, which is independent of the task instruction. Therefore, whether a model achieves a high score is determined primarily by its own understanding of physical tools and its reasoning capabilities. We also included selected quantitative results in the rebuttal to further substantiate this claim.
>
> For other issues not mentioned here and more details, please kindly refer to our detailed rebuttal response.
>
> **We sincerely hope this addresses your concerns, and we very much look forward to any further discussion.**
>
> Warm regards,
>
> Authors

---

### Author Response · Authors · 2025-12-01
**Summary of the Paper, Reviews, Rebuttal and Discussion**

**Dear Area Chair,**

To save your valuable time, we provide an objective summary of the paper, reviews, rebuttal, and discussion below.

## Summary of Our Paper in Three Sentences

We introduce PhysToolBench, the **first benchmark** designed to **systematically evaluate MLLMs’ understanding of physical tools** through three progressively challenging levels.

We conducted an **extensive evaluation** of 32 MLLMs across four major categories and provided an **in-depth analysis**.

We also propose a simple **baseline**, Vision-Centric Reasoning (VCR), demonstrating the crucial role of visual reasoning in tool understanding and providing insights for future research.

## Summary of Reviews

The paper's **novelty, contribution, and writing quality** are well recognized.
*   **Novelty:** Reviewers (**FTcy, RBRq, nNjs**) acknowledge PhysToolBench as a novel benchmark for the unexplored problem of physical tool understanding, well reflecting embodied settings.
*   **Experiments:** **All reviewers** praise the extensive evaluation of 32 MLLMs and the detailed analysis provided.
*   **Clarity:** The paper is recognized as well-written and easy to follow (**FTcy, RBRq, nNjs**).

However, there are still some misunderstandings and concerns, which mainly fell into three categories: benchmark design, VCR baseline details, and ICLR suitability.

## Summary of Rebuttal & Discussion

**We responded to each misunderstanding and concern in the rebuttal and revision with clarifications and illustrative cases.**
FTcy confirmed all concerns were fully addressed, but we have not received any responses from the other three reviewers.

### With 6cbv:
*   **Suitability for ICLR:** 6cbv expressed concern that our paper might not be suitable for ICLR. We highlighted that the ICLR main conference frequently accepts benchmark-related works in our rebuttal.
*   **GPT Bias:** 6cbv questioned whether the GPT series gains an unfair advantage because some images were generated by GPT-4o. We clarified that GPT-4o generates images solely based on *scene descriptions*, which are independent of *task descriptions*. Therefore, a model's success depends on its understanding of physical tools and reasoning capabilities. We also provided data to support this.

**We sincerely believe these concerns do not diminish our contribution**, especially the concern regarding suitability for ICLR, which we believe is somewhat unwarranted.

### With FTcy:
*   **Distinguishability of MLLMs' Performance:** FTcy questioned whether our benchmark effectively distinguishes the performance of different MLLMs. **We added detailed statistical data** (including max, min, mean, and standard deviation of MLLMs' performance for each subset), **demonstrating that our benchmark significantly differentiates between models**.
*   **Cases about Tool Material:** FTcy was curious about how MLLMs perform on cases involving tools with identical appearances but different materials. Our benchmark **actually includes such cases**, and we added further clarification and some examples in the revised manuscript to highlight this.

During discussion, FTcy confirmed that **all concerns have been resolved.**

### With RBRq:
*   **Potential Misunderstanding on Benchmark Design:** RBRq initially had potential misunderstandings regarding the benchmark design, which led to incorrect criticisms. We provided a thorough clarification in the rebuttal and added plenty of representative cases for each misunderstanding in the manuscript revision to address this.
*   **Suitability for Main Conference:** RBRq suggested submitting to "benchmark track" (**which does not exist at ICLR**). We clarified in the rebuttal with examples that the ICLR main conference is an appropriate and common venue for benchmark papers.

Although RBRq repeatedly acknowledged our novelty and contribution, the initial low score (4) was based on these misunderstandings. We tried to address these misunderstandings in the rebuttal, but no response has been received.

### With nNjS:
*   **Real-World Transfer:** nNjS expressed concern regarding the real-world transferability of the benchmark. In the rebuttal, we compared results across different subsets (real-world vs. generated) and found that performance differences are minor, suggesting our benchmark does not suffer from a significant domain gap.
*   **Potential Misunderstanding on Ambiguity:** nNjS noted ambiguity in the benchmark design. We clarified how we ensure minimal ambiguity and added numerous representative examples in the revision.
*   **More Details about VCR:** nNjS showed interest in our baseline method (VCR), noting it is promising but requested more results and details. Although the method is not the paper's primary focus, we appreciated the rigorous suggestion and provided relevant results and technical details in both the rebuttal and revision.

We appreciate the suggestions and made thorough clarifications and revisions, yet no response has been received.

---

### Meta-Review · Area_Chair_gj13 · 2026-01-10

**Summary:**

The main concerns from the reviewers are following:

- Reviewer **6cbv**:
  - **W1**:  Images are generated by GPT-4o, leading to the potential privilege for GPT family models.

- Reviewer **RBRq**:
  - **W1**: The benchmark's scope is thus limited in its ability to truly test the comprehension of physics.
  - **W2**: The paper does not address how the benchmark handles the ambiguity in tool creation.
  - **W3**: The provided images appear to be heavily staged and simplified compared with real open-world environments.

- Reviewer **nNjS**:
  - **W1**: Most images are generated by GPT-4o rather than captured in real environments.
  - **W2**: For M2 Tool Combination, success may depend on understanding hidden state not visually disambiguated.
  - **W3**: The proposed VCR pipeline is only evaluated on M3 with two backbones (GPT-4o/5).
  - **W4**: How to treat cases with multiple acceptable substitutes or partially adequate tools

Furthermore, multiple Reviewers (**6cbv, RBRq, nNjS**) express their concerns that the paper focuses only on proposing a benchmark and conducting the corresponding experimental analysis on existing models, rather than proposing a technical contribution.

**Reviewer Concerns:**

The author responses include fruitful discussions and additional experimental results, while there are still concerns remained outstanding:

  - A major concern lies in that most images are generated by GPT-4o rather than captured in real environments (**W1** of Reviewer **6cbv**, **W1** of Reviewer **nNjS**). Even though authors have provided additional results to show that the difference of performance is minor for tasks with LLM-generated and real images, the concerns regarding whether the benchmark captures real-world challenging scenarios (**W2, W3** of Reviewer **RBRq**, **W2, W4** of Reviewer **nNjS**) are more significant due to this. The benchmark can be significantly strengthened with more real-world captured scenes.

  - Even though benchmark contribution can also be recognized in ICLR, I agree with the Reviewers (**6cbv, RBRq, nNjS**) that the contribution of the paper can be more significant if more technical contributions are included, such as proposing an agent training method as suggested by Reviewer **nNjS**.

**Reviewer Scores:**

As discussed in the Reviewer Concerns, some points are indeed addressed by the author responses. While in my view, the remaining concerns (**W1** of Reviewer **6cbv**, **W2, W3** of Reviewer **RBRq**, **W1, W2, W4** of Reviewer **nNjS**, the concern of proposing benchmark only) also prevent the reviewers with negative evaluations to fully change their opinions.

---

### Decision · Program_Chairs · 2026-01-26

Reject